**Trends in secondary inorganic aerosol pollution in China and its responses to emission controls of precursors in wintertime**

Fanlei Meng[1#], Yibo Zhang[2#], Jiahui Kang[1], Mathew R. Heal[3], Stefan Reis[4,3,5], Mengru Wang[6], Lei Liu[7], Kai Wang[1], Shaocai Yu[2*], Pengfei Li[8], Jing Wei[9], Yong Hou[1], Ying Zhang[1], Xuejun Liu[1], Zhenling Cui[1], Wen Xu[1*], Fusuo Zhang[1]

[1]College of Resources and Environmental Sciences, National Academy of Agriculture Green Development, Key Laboratory of Plant–Soil Interactions, Ministry of Education, National Observation and Research Station of Agriculture Green Development (Quzhou, Hebei), China Agricultural University, Beijing 100193, China.
[2]Research Center for Air Pollution and Health, Key Laboratory of Environmental Remediation and Ecological Health, Ministry of Education, College of Environment and Resource Sciences, Zhejiang University, Hangzhou, Zhejiang 310058, P.R. China
[3]School of Chemistry, The University of Edinburgh, David Brewster Road, Edinburgh EH9 3FJ, United Kingdom
[4]UK Centre for Ecology & Hydrology, Penicuik, EH26 0QB, United Kingdom.
[5]University of Exeter Medical School, Knowledge Spa, Truro, TR1 3HD United Kingdom.
[6]Water Systems and Global Change Group, Wageningen University & Research, P.O. Box 47, 6700 AA Wageningen, The Netherlands
[7]College of Earth and Environmental Sciences, Lanzhou University, Lanzhou 730000, China
[8]College of Science and Technology, Hebei Agricultural University, Baoding, Hebei 071000, China
[9]Department of Atmospheric and Oceanic Science, Earth System Science Interdisciplinary Center, University of Maryland, College Park 20740, USA

*Corresponding authors

E-mail addresses: W. Xu (wenxu@cau.edu.cn ); S C. Yu (shaocaiyu@zju.edu.cn )

[#] Contributed equally to this work.

**ABSTRACT:** The Chinese government recently proposed ammonia ($NH_3$) emissions reductions (but without a specific national target) as a strategic option to mitigate $PM_{2.5}$ pollution. We combined a meta-analysis of nationwide measurements and air quality modelling to identify efficiency gains by striking a balance between controlling $NH_3$ and acid gas ($SO_2$ and $NO_x$) emissions. We found that $PM_{2.5}$ concentrations decreased from 2000 to 2019, but annual mean $PM_{2.5}$ concentrations still exceeded 35 µg m$^{-3}$ at 74% of 1498 monitoring sites in 2015-2019. The concentration of $PM_{2.5}$ and its components were significantly higher (16%-195%) on hazy days than on non-hazy days. Compared with mean values of other components, this difference was more significant for the secondary inorganic ions $SO_4^{2-}$, $NO_3^-$, and $NH_4^+$ (average increase 98%). While sulfate concentrations significantly decreased over the time period, no significant change was observed for nitrate and ammonium concentrations. Model simulations indicate that the effectiveness of a 50% $NH_3$ emission reduction for controlling SIA concentrations decreased from 2010 to 2017 in four megacity clusters of eastern China, simulated for the month of January under fixed meteorological conditions (2010). Although the effectiveness further declined in 2020 for simulations including the natural experiment of substantial reductions in acid gas emissions during the COVID-19 pandemic, the resulting reductions in SIA concentrations were on average 20.8% lower than that in 2017. In addition, the reduction of SIA concentrations in 2017 was greater for 50% acid gas reductions than for the 50% $NH_3$ emissions reduction. Our findings indicate that persistent secondary inorganic aerosol pollution in China is limited by acid gases emissions, while an additional control on $NH_3$ emissions would become more important as reductions of $SO_2$ and $NO_x$ emissions progress.

56

**Keywords:** Air pollution, Particulate matter, Second inorganic aerosols, Anthropogenic emission, Ammonia.

## 1. Introduction

Over the past two decades, China has experienced severe $PM_{2.5}$ (particulate matter with aerodynamic diameter $\leq$ 2.5 μm) pollution (Huang et al., 2014; Wang et al., 2016), leading to adverse impacts on human health (Liang et al., 2020) and the environment (Yue et al., 2020). In 2019, elevated $PM_{2.5}$ concentrations accounted for 46% of polluted days in China and $PM_{2.5}$ was officially identified as a key year-round air pollutant (MEEP, 2019). Mitigation of $PM_{2.5}$ pollution is therefore the most pressing current challenge to improve China's air quality.

The Chinese government has put a major focus on particulate air pollution control through a series of policies, regulations, and laws to prevent and control severe air pollution. Before 2010, the Chinese government mainly focused on controlling $SO_2$ emissions via improvement of energy efficiency, with less attention paid to $NO_x$ abatement (CSC, 2007, 2011, 2016). For example, the 11[th] Five-Year Plan (FYP) (2006-2010) set a binding goal of a 10% reduction for $SO_2$ emission (CSC, 2007). The 12[th] FYP (2011-2015) added $NO_x$ regulation and required 8% and 10% reductions for $SO_2$ and $NO_x$ emissions, respectively (CSC, 2011) This was followed by further reductions in $SO_2$ and $NO_x$ emissions of 15% and 10%, respectively, in the 13[th] FYP (2016-2020) (CSC, 2016). In response to the severe haze events of 2013, the Chinese State Council promulgated the toughest-ever 'Atmospheric Pollution Prevention and Control Action Plan' in September 2013, aiming to reduce ambient $PM_{2.5}$ concentrations by 15-20% in 2017 relative to 2013 levels in metropolitan regions (CSC, 2013). As a result of the implementation of stringent control measures, emissions reductions markedly

accelerated from 2013-2017, with decreases of 59% for $SO_2$, 21% for $NO_x$, and 33%

for primary $PM_{2.5}$ (Zheng et al., 2018). Consequently, significant reductions in annual

mean $PM_{2.5}$ concentrations were observed nationwide (Zhang et al., 2019; Yue et al.,

2020), in the range 28-40% in the metropolitan regions (CSC, 2018a). To continue its

efforts in tackling air pollution, China promulgated the Three-Year Action Plan (TYAP)

in 2018 for Winning the Blue-Sky Defense Battle (CSC, 2018b), which required a

further 15% reduction in $NO_x$ emissions by 2020 compared to 2018 levels.

Despite a substantial reduction in $PM_{2.5}$ concentrations in China, the proportion of

secondary aerosols during severe haze periods is increasing (An et al., 2019), and can

comprise up to 70% of $PM_{2.5}$ concentrations (Huang et al., 2014). Secondary inorganic

aerosols (SIA, the sum of sulfate ($SO_4^{2-}$), nitrate ($NO_3^-$), and ammonium ($NH_4^+$)) were

found to be of equal importance to secondary organic aerosols, with 40-50%

contributions to $PM_{2.5}$ in eastern China (Huang et al., 2014; Yang et al., 2011). The acid

gases (i.e., $NO_x$, $SO_2$), together with $NH_3$, are crucial precursors of SIA via chemical

reactions that form particulate ammonium sulfate, ammonium bisulfate, and

ammonium nitrate (Ianniello et al., 2010). In addition to the adverse impacts on human

health via fine particulate matter formation (Liang et al., 2020; Kuerban et al., 2020),

large amounts of $NH_3$ and its aerosol-phase products also lead to nitrogen deposition

and consequently to environmental degradation (Ortiz-Montalvo et al., 2014; Pan et al.,

2015; Xu et al., 2015, 2018; Zhan et al., 2021).

Following the successful controls on $NO_x$ and $SO_2$ emissions since 2013 in China,

some studies found $SO_4^{2-}$ exhibited much larger decline than $NO_3^-$ and $NH_4^+$ , which

lead to a rapid transition from sulfate-driven to nitrate-driven aerosol pollution (Li et

al., 2019, 2021; Zhang et al., 2019). Attention is turning to $NH_3$ emissions as a possible

means of further $PM_{2.5}$ control (Bai et al., 2019; Kang et al., 2016), particularly as

emissions of $NH_3$ increased between the 1980s and 2010s. Some studies have found that $NH_3$ limited the formation of SIA in winter in the eastern United States (Pinder et al., 2007) and Europe (Megaritis et al., 2013). Controls on $NH_3$ emissions have been proposed in the TYAP, although mandatory measures and binding targets have not yet been set (CSC, 2018b). Nevertheless, this proposal means that China will enter a new phase of $PM_{2.5}$ mitigation, with attention now given to both acid gas and $NH_3$ emissions. However, in the context of effective control of $PM_{2.5}$ pollution via its SIA component, two key questions arise: 1) what are the responses of the constituents of SIA to implementation of air pollution control policies, and 2) what is the relative efficiency of $NH_3$ versus acid gas emission controls to reduce SIA pollution?

To fill this evidence gap and provide useful insights for policy-making to improve air quality in China, this study adopts an integrated assessment framework. With respect to the emission control policy summarized above, China's $PM_{2.5}$ control can be divided into three periods: period I (2000–2012), in which $PM_{2.5}$ was not the targeted pollutant; period II (2013–2016), the early stage of targeted $PM_{2.5}$ control policy implementation; and period III (2017–2019), the latter stage with more stringent policies. Therefore, our research framework consists of two parts: (1) assessment of trends in annual mean concentrations of $PM_{2.5}$, its chemical components and SIA gaseous precursors from meta-analyses and observations; (2) quantification of SIA responses to emissions reductions in $NH_3$ and acid gases using the Weather Research and Forecasting and Community Multiscale Air Quality (WRF/CMAQ) models.

## 2. Materials and methods

### 2.1. Research framework

This study developed an integrated assessment framework to analysis the trends of secondary inorganic aerosol and strategic options to reduce SIA and $PM_{2.5}$ pollution in

China (Fig. 1). The difference in PM$_{2.5}$ chemical components between hazy and non-hazy days was first assessed by meta-analysis of published studies. These were interpreted in conjunction with the trends in air concentrations of PM$_{2.5}$ and its secondary inorganic aerosol precursors (SO$_2$, NO$_2$, and NH$_3$) derived from surface measurements and satellite observations. The potential of SIA and PM$_{2.5}$ concentration reductions from precursor emission reductions was then evaluated using the Weather Research and Forecasting and Community Multiscale Air Quality (WRF/CMAQ) models.

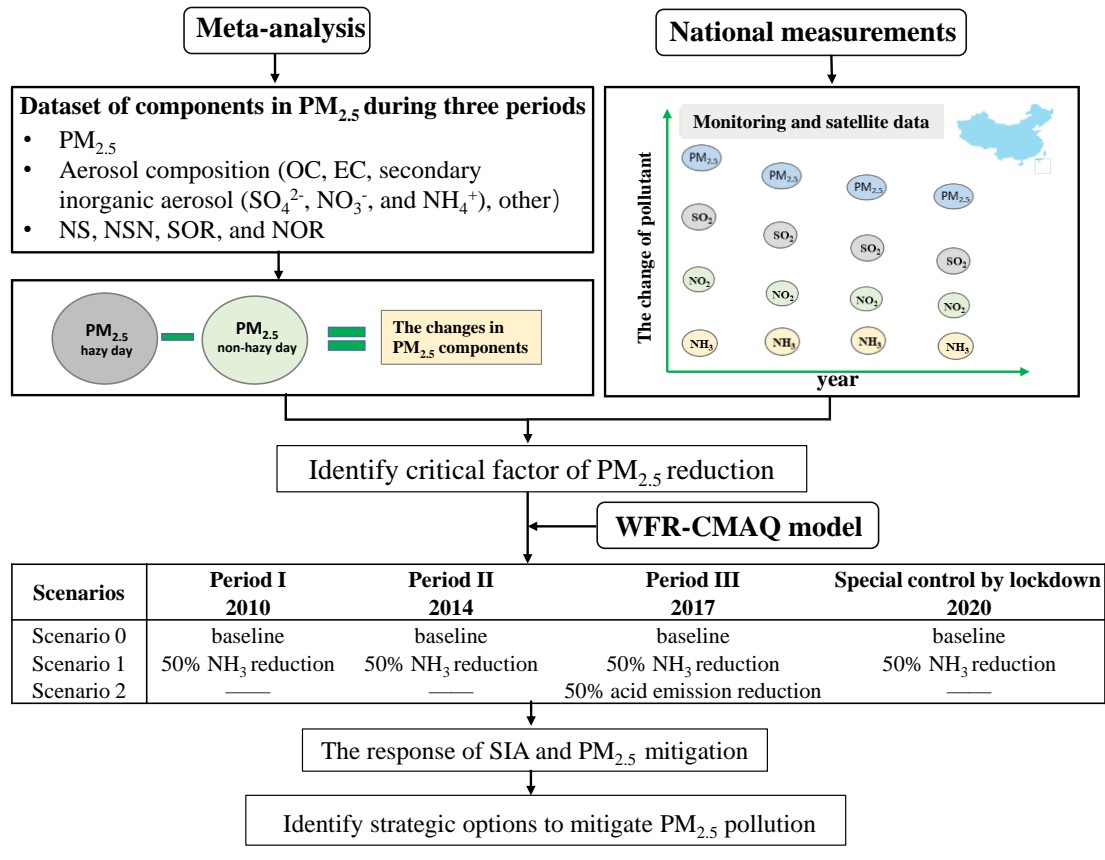

**Fig. 1.** Integrated assessment framework for Chinese PM$_{2.5}$ mitigation strategic options. OC is organic carbon, EC is elemental carbon, NO$_3^-$ is nitrate, SO$_4^{2-}$ is sulfate, and NH$_4^+$ is ammonium. NS is the slope of the regression equation between [NH$_4^+$] and [SO$_4^{2-}$], NSN is the slope of the regression equation between [NH$_4^+$] and [SO$_4^{2-}$ + NO$_3^-$], SOR is sulfur oxidation ratio, and NOR is nitrogen oxidation ratio. SIA is secondary

inorganic aerosols. WRF-CMAQ is Weather Research and Forecasting and Community

Multiscale Air Quality models.

**2.2. Meta-analysis of $PM_{2.5}$ and its chemical components**

Meta-analyses can be used to quantify the differences in concentrations of $PM_{2.5}$ and

its secondary inorganic aerosol components ($NH_4^+$, $NO_3^-$, and $SO_4^{2-}$) between hazy and

non-hazy days and to identify the major pollutants on non-hazy days (Wang et al.,

2019b); this provides evidence for effective options on control of precursor emissions

($NH_3$, $NO_2$, and $SO_2$) for reducing occurrences of hazy days. To build a database of

atmospheric concentrations of $PM_{2.5}$ and chemical components between hazy and non-

hazy days, we conducted a literature survey using the Web of Science and the China

National Knowledge Infrastructure for papers published between January 2000 and

January 2020. The keywords included: (1) "particulate matter," or "aerosol," or "$PM_{2.5}$"

and (2) "China" or "Chinese". Studies were selected based on the following conditions:

(1) Measurements were taken on both hazy and non-hazy days.

(2) $PM_{2.5}$ chemical components were reported.

(3) If hazy days were not defined in the screened articles, the days with $PM_{2.5}$

concentrations $> 75$ μg m$^{-3}$ (the Chinese Ambient Air Quality Standard Grade II for

$PM_{2.5}$ (CSC, 2012)) were treated as hazy days.

(4) If an article reported measurements from different monitoring sites in the same city,

e.g. Mao et al. (2018) and Xu et al. (2019), then each measurement was considered an

independent study.

(5) If there were measurements in the same city for the same year, e.g. Tao et al. (2016)

and Han et al. (2017), then each measurement was treated as an independent study.

One hundred articles were selected based on the above conditions with the lists

provided in the Supporting Material dataset. For each selected study, we documented

the study sites, study periods, seasons, aerosol types, and aerosol species mass concentrations (in µg m$^{-3}$) over the entire study period (2000–2019) (the detailed data are provided in the dataset). In total, the number of sites contributing data to the meta-analysis was 267 and their locations are shown in Fig. S1. If relevant data were not directly presented in studies, a GetData Graph Digitizer (Version 2.25, http://www.getdatagraph-digitizer.com) was used to digitize concentrations of PM$_{2.5}$ chemical components from figures. The derivations of other variables such as sulfur and nitrogen oxidation ratios are described in Supplementary Information Method 1.

Effect sizes were developed to normalize the combined studies' outcomes to the same scale. This was done through the use of log response ratios (lnRR) (Nakagawa et al., 2012; Ying et al., 2019). The variations in aerosol species were evaluated as follows:

$$\ln RR = \ln \left(\frac{X_p}{X_n}\right) \qquad (1)$$

where $X_p$ and $X_n$ represent the mean values of the studied variables of PM$_{2.5}$ components on hazy and non-hazy days, respectively. The mean response ratio was then estimated as:

$$RR = \exp \left[\sum \ln RR(i) \times W(i) / \sum W(i) \right] \qquad (2)$$

where $W(i)$ is the weight given to that observation as described below. Finally, variable-related effects were expressed as percent changes, calculated as (RR−1) ×100%. A 95% confidence interval not overlapping with zero indicates that the difference is significant. A positive or negative percentage value indicates an increase or decrease in the response variables, respectively.

We used inverse sampling variances to weight the observed effect size (RR) in the meta-analysis (Benitez-Lopez et al., 2017). For the measurement sites where standard deviations (SD) or standard errors (SE) were absent in the original study reports, we used the "Bracken, 1992" approach to estimate SD (Bracken et al., 1992). The variation-

related chemical composition of PM$_{2.5}$ was assessed by random effects in meta-analysis. Rosenberg's fail safe-numbers ($N_{f_s}$) were calculated to assess the robustness of findings on PM$_{2.5}$ to publication bias (Ying et al., 2019) (See Table S1). The results (effects) were considered robust despite the possibility of publication bias if $N_{f_s} > 5 \times n + 10$, where $n$ indicates the number of sites. The statistical analysis of the concentrations of PM$_{2.5}$ and secondary inorganic ions for three periods used a non-parametric statistical method since concentrations were not normally distributed based on the Kruskal-Wallis test (Kruskal and Walls, 1952). For each species, the Kruskal-Wallis one-way analysis of variance (ANOVA) on ranks among three periods was performed with pairwise comparison using Dunn's method (Dunn, 1964).

**2.3. Data collection of air pollutant concentrations**

To assess the recent annual trends in China of PM$_{2.5}$ and of the SO$_2$ and NO$_2$ gaseous precursors to SIA, real-time monitoring data of these pollutants at 1498 monitoring stations in 367 cities during 2015–2019 were obtained from the China National Environmental Monitoring Center (CNEMC) (http://106.37.208.233:20035/). This is an open-access archive of air pollutant measurements from all prefecture-level cities since January 2015. Successful use of data from CNEMC to determine characteristics of air pollution and related health risks in China has been demonstrated previously (Liu et al., 2016; Kuerban et al., 2020). The geography stations are shown in Fig. S1. The annual mean concentrations of the three pollutants at all sites were calculated from the hourly time-series data according to the method of Kuerban et al. (2020). Information about sampling instruments, sampling methods, and data quality controls for PM$_{2.5}$, SO$_2$, and NO$_2$ is provided in Supplementary Method 2. Surface NH$_3$ concentrations over China for the 2008–2016 (the currently available) were extracted from the study of Liu et al. (2019a). Further details are in Supplementary Method 2.

**2.4. WRF/CMAQ model simulations**

The Weather Research and Forecasting model (WRFv3.8) and the Models-3 community multi-scale air quality (CMAQv5.2) model were used to evaluate the impacts of emission reductions on SIA and $PM_{2.5}$ concentrations over China. The simulations were conducted at a horizontal resolution of 12 km × 12 km. The simulation domain covered the whole of China, part of India and east Asia. In the current study, focus was on the following four regions in China: Beijing-Tianjin-Hebei (BTH), Yangtze River Delta (YRD), Pearl River Delta (PRD), and Sichuan Basin (SCB). The model configurations used in this study were the same as those used in Wu et al. (2018a) and are briefly described here. The WRFv3.8 model was applied to generate meteorological inputs for the CMAQ model using the National Center for Environmental Prediction Final Operational Global Analysis (NCEP-FNL) dataset (Morrison et al., 2009). Default initial and boundary conditions were used in the simulations. The carbon-bond (CB05) gas-phase chemical mechanism and AERO6 aerosol module were selected in the CMAQ configuration (Guenther et al., 2012). Anthropogenic emissions for 2010, 2014 and 2017 were obtained from the Multi-resolution Emission Inventory (http://meicmodel.org) with $0.25° × 0.25°$ spatial resolution and aggregated to 12 km×12 km resolution (Zheng et al., 2018; Li et al., 2017). Each simulation was spun-up for six days in advance to eliminate the effects of the initial conditions.

The years 2010, 2014 and 2017 were chosen to represent the anthropogenic emissions associated with the periods I, II, III, respectively. January was selected as the typical simulation month because wintertime haze pollution frequently occurs in this month (Wang et al., 2011; Liu et al., 2019b). January of 2010 was also found to have $PM_{2.5}$ pollution more serious than other months (Geng et al., 2017, 2021). The

sensitivity scenarios of emissions in January can therefore help to identify the efficient option to control haze pollution.

The Chinese government has put a major focus on acid gas emission control through a series of policies in the past three periods (Fig. S2). The ratio decreases of anthropogenic emissions $SO_2$ and $NO_x$ in January for the years 2010, 2014, 2017 and 2020 are presented in SI Tables S2 and S3, respectively. The emissions from surrounding countries were obtained from the Emissions Database for Global Atmospheric Research (EDGAR): HTAPV2. The scenarios and the associated reductions of $NH_3$, $NO_x$ and $SO_2$ for selected four years in three periods can be found in Fig. 1.

The sensitivities of SIA and $PM_{2.5}$ to $NH_3$ emissions reductions were determined from the average $PM_{2.5}$ concentrations in model simulations without and with an additional 50% $NH_3$ emissions reduction. The choice of 50% additional $NH_3$ emissions reduction is based on the feasibility and current upper bound of $NH_3$ emissions reduction expected to be realized in the near future (Liu et al., 2019a; Zhang et al., 2020a; Table S4). For example, Zhang et al. (2020a) found that the mitigation potential of $NH_3$ emissions from cropland production and livestock production in China can reach up to 52% and 58%, respectively. To eliminate the influences of varying meteorological conditions, all simulations were conducted under the fixed meteorological conditions of 2010.

During the COVID-19 lockdown in China, emissions of primary pollutants were subject to unprecedented reductions due to national restrictions on traffic and industry; in particular, emissions of $NO_x$ and $SO_2$ reduced by 46% and 24%, respectively, averaged across all Chinese provinces (Huang et al., 2021). We therefore also ran simulations applying the same reductions in $NO_x$ and $SO_2$ (based on 2017 MEIC) that

were actually observed during the COVID-19 lockdown as a case of special control in 2020.

**2.5 Model performance**

The CMAQ model has been extensively used in air quality studies (Zhang et al., 2019; Backes et al., 2016) and the validity of the chemical regime in the CMAQ model had been confirmed by our previous studies (Zhang et al., 2021a; Wang et al., 2020a, 2021a). In this study, we used surface measurements from previous publications (e.g., (Xiao et al., 2020, 2021; Geng et al., 2019; Xue et al., 2019) and satellite observations to validate the modelling meteorological parameters by WRF model and air concentrations of $PM_{2.5}$ and associated chemical components by CMAQ model. The meteorological measurements used for validating the WRF model performances were obtained from the National Climate Data Center (NCDC) (ftp://ftp.ncdc.noaa.gov/pub/data/noaa/). For validation of the CMAQ model, monthly mean concentrations of $PM_{2.5}$ were obtained from ChinaHighAirPollutants (CHAP, https://weijing-rs.github.io/product.html) database. We also collected ground-based observations from previous publications to validate the modeling concentrations of $SO_4^{2-}$, $NO_3^-$, and $NH_4^+$. The detailed information of the monitoring sites is presented in Table S5. Further information about the modelling is given in Supplementary Method 3 and Figs. S3-S7 and Table S5.

**3. Results and discussion**

**3.1. Characteristics of $PM_{2.5}$ and its chemical components from the meta-analysis and from nationwide observations**

The meta-analysis based on all published analyses of $PM_{2.5}$ and chemical component measurements during 2000–2019 reveals the changing characteristics of $PM_{2.5}$. To assess the annual trends in $PM_{2.5}$ and its major chemical components, we

made a three-period comparison using the measurements at sites that include both $PM_{2.5}$
and secondary inorganic ions $SO_4^{2-}$, $NO_3^-$, and $NH_4^+$ (Fig. 2). The $PM_{2.5}$ concentrations
on both hazy and non-hazy days showed no significant trend from period I to period II
based on the Kruskal-Wallis test. This could be explained by the enhanced atmospheric
oxidation capacity (Huang et al., 2021), faster deposition of total inorganic nitrate (Zhai
et al., 2021) and the changes of atmospheric circulation (Zheng et al., 2015; Li et al.,
2020). However, the observed concentrations of $PM_{2.5}$ showed a downward trend from
Period I to Period III on the non-hazy days, decreasing by 8.2% (Fig. 2a), despite no
significant decreasing trend on the hazy days (Fig. 2a). In addition, the annual mean
$PM_{2.5}$ concentrations from the nationwide measurements showed declining trends
during 2015-2019 averaged across all China and for each of the BTH, YRD, SCB, and
PRD megacity clusters of eastern China (Fig. 3a, d).
These results reflect the effectiveness of the pollution control policies (Fig. S2)
implemented by the Chinese government at the national scale. Nevertheless, $PM_{2.5}$
remained at relatively high levels. Over 2015–2019, the annual mean $PM_{2.5}$
concentrations at 74% of the 1498 sites (averaging $51.9 \pm 12.4 \ \mu g \ m^{-3}$, Fig. 3a) exceeded
the Chinese Grade-II Standard (GB 3095–2012) of 35 $\mu g \ m^{-3}$ (MEPC, 2012), indicating
that $PM_{2.5}$ mitigation is a significant challenge for China.

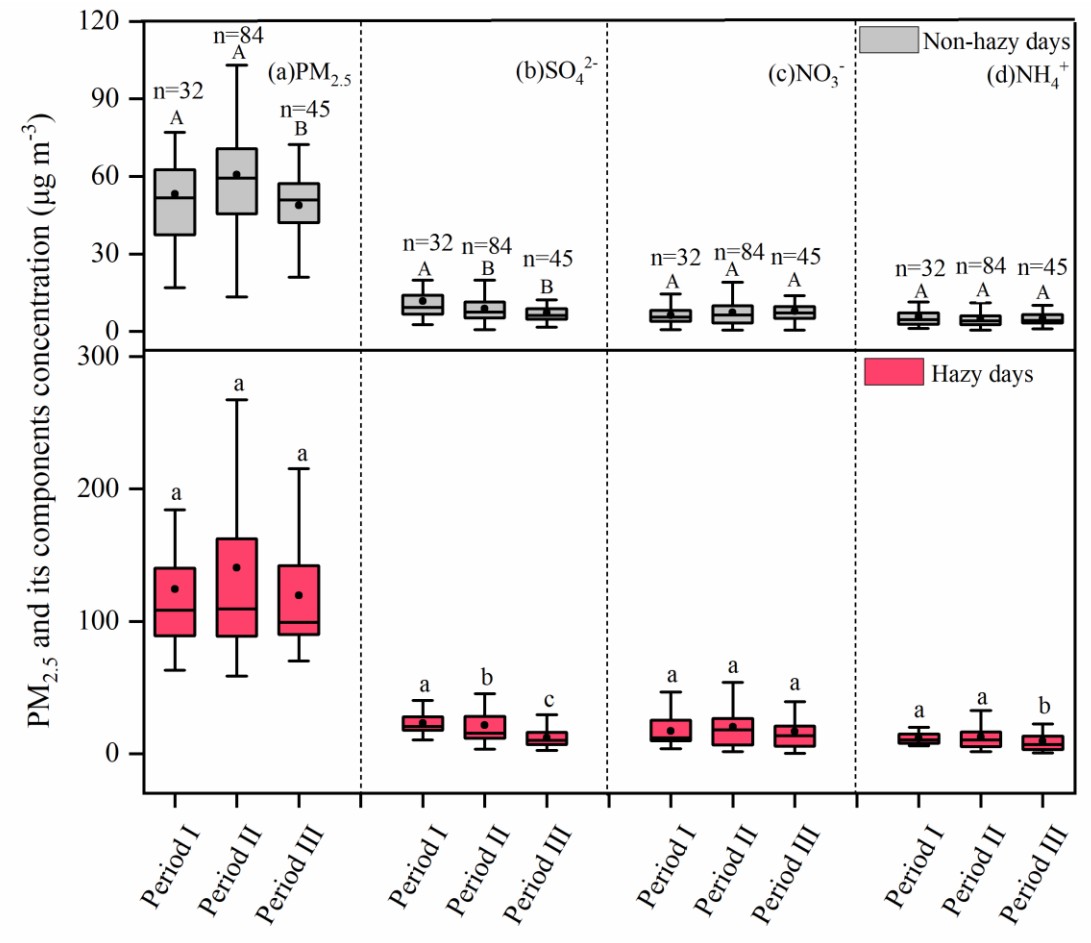

**Fig. 2.** Comparisons of observed concentrations of (a) $PM_{2.5}$, (b) $SO_4^{2-}$, (c) $NO_3^-$, and (d) $NH_4^+$ between non-hazy and hazy days in Period I (2000–2012), Period II (2013–2016), and Period III (2017–2019). Bars with different letters denote significant differences among the three periods ($P < 0.05$) (upper and lowercase letters for non-hazy and hazy days, respectively). The upper and lower boundaries of the boxes represent the 75th and 25th percentiles; the line within the box represents the median value; the whiskers above and below the boxes represent the 90th and 10th percentiles; the point within the box represents the mean value. Comparison of the pollutants among the three-periods using Kruskal-Wallis and Dunn's test. The $n$ represents independent sites; more detail on this is presented in Section 2.2.

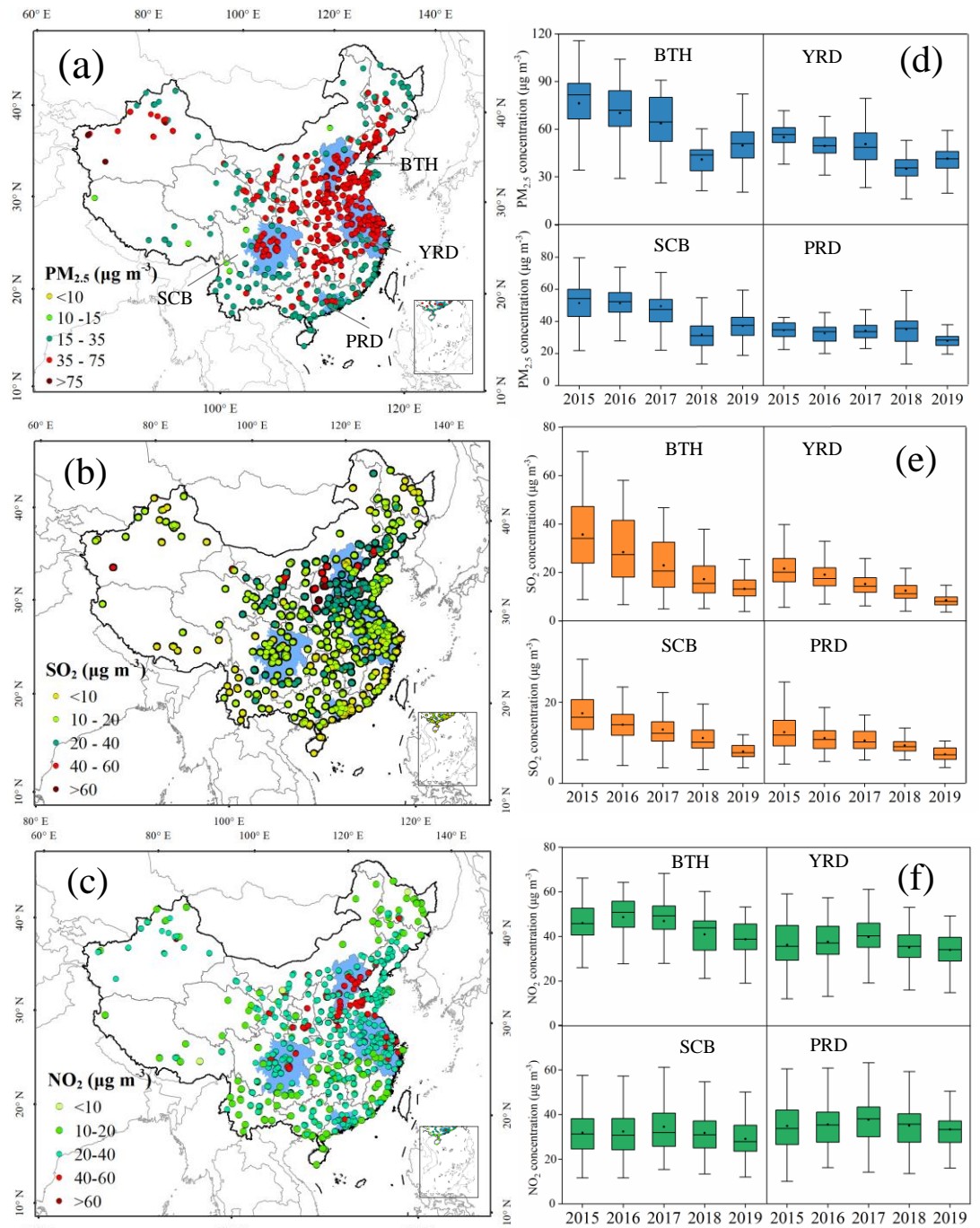

**Fig. 3.** Left: spatial patterns of annual mean observed concentration of (a) $PM_{2.5}$, (b) $SO_2$, (c) $NO_2$ at 1498 sites, averaged for 2015–2019. Right: the annual observed concentrations of (d) $PM_{2.5}$, (e) $SO_2$, and (f) $NO_2$ for 2015-2019 in four megacity clusters (BTH: Beijing-Tianjin-Hebei, YRD: Yangtze River Delta, SCB: Sichuan Basin, PRD: Pearl River Delta). The locations of the regions are indicated by the blue shading on the map. The upper and lower boundaries of the boxes represent the 75th and 25th

percentiles; the line within the box represents the median value; the whiskers above and
below the boxes represent the 90th and 10th percentiles; the point within the box
represents the mean value.
To further explore the underlying drivers of $PM_{2.5}$ pollution, we analyzed the
characteristics of $PM_{2.5}$ chemical components and their temporal changes in China. The
concentrations of $PM_{2.5}$ and all its chemical components (except $F^-$ and $Ca^{2+}$) were
significantly higher on hazy days than on non-hazy days (Fig. 4A). Compared with
other components this difference was more significant for secondary inorganic ions (i.e.,
$SO_4^{2-}$, $NO_3^-$, and $NH_4^+$). Sulfur oxidation ratio (SOR) and nitrogen oxidation ratio
(NOR) were also 58.0% and 94.4% higher on hazy days than on non-hazy days,
respectively, implying higher oxidations of gaseous species to sulfate- and nitrate-
containing aerosols on the hazy days (Sun et al., 2006; Xu et al., 2017).
To provide quantitative information on differences in $PM_{2.5}$ and its components
between hazy days and non-hazy days, we made a comparison using 46 groups of data
on simultaneous measurements of $PM_{2.5}$ and chemical components. The 46 groups refer
to independent analyses from the literature that compare concentrations of $PM_{2.5}$ and
major components ($SO_4^{2-}$, $NO_3^-$, $NH_4^+$, OC, and EC) on hazy and non-hazy days
measured across different sets of sites. The "Other" species was calculated by
difference between $PM_{2.5}$ and sum of OC, EC, and secondary inorganic ions ($SO_4^{2-}$,
$NO_3^-$ and $NH_4^+$). As shown in Fig.4B (a), $PM_{2.5}$ concentrations significantly increased
(by 136%) on the hazy days ($149.2 \pm 81.6$ μg m$^{-3}$) relative to those on the non-hazy
days ($63.2 \pm 29.8$ μg m$^{-3}$). By contrast, each component's proportions within $PM_{2.5}$
differed slightly, with 36% and 40% contributions by SIA on non-hazy days and hazy
days, respectively (Fig. 4B(b)). This is not surprising because concentrations of $PM_{2.5}$
and SIA both significantly increased on the hazy days ($60.1 \pm 37.4$ μg m$^{-3}$ for SIA)

relative to the non-hazy days (22.4 ± 12.1 µg m⁻³ for SIA). Previous studies have found that increased SIA formation is the major influencing factor for haze pollution in wintertime and summertime (mainly in years since 2013) in major Chinese cities in eastern China (Huang et al., 2014; Wang et al., 2019a; Li et al., 2018). Our results extend confirmation of the dominant role of SIA to PM$_{2.5}$ pollution over a large spatial scale in China and to longer temporal scales.

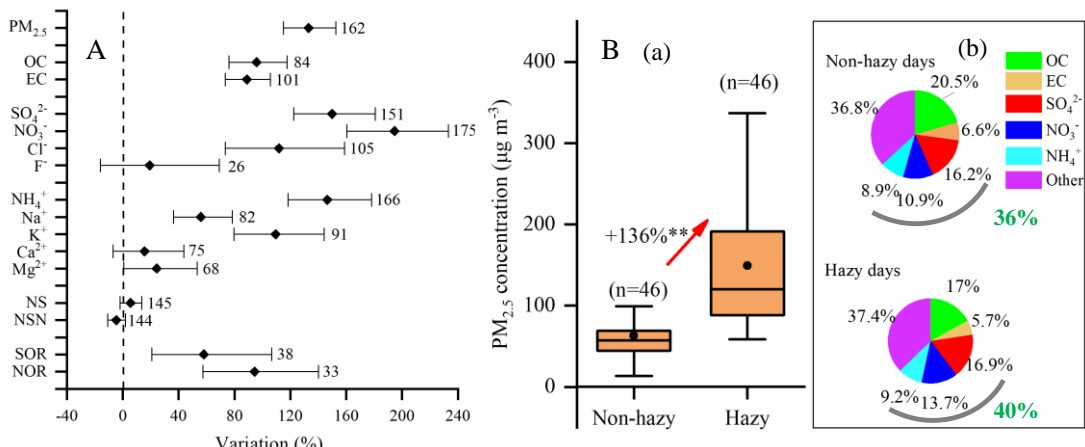

**Fig. 4.** (A) Variations in PM$_{2.5}$ concentration, aerosol component concentration, NS, NSN, SOR, and NOR from non-hazy to hazy days in China during 2000–2019. (B) (a) Summary of differences in PM$_{2.5}$ concentration between non-hazy and hazy days in China; (b) the average proportions of components of PM$_{2.5}$ on non-hazy and hazy days. NS is the slope of the regression equation between [NH$_4^+$] and [SO$_4^{2-}$], NSN is the slope of the regression equation between [NH$_4^+$] and [SO$_4^{2-}$ + NO$_3^-$], SOR is sulfur oxidation ratio, and NOR is nitrogen oxidation ratio. The variations are considered significant if the confidence intervals of the effect size do not overlap with zero. [**] denotes significant difference ($P$ <0.01) between hazy days and non-hazy days. The upper and lower boundaries of the boxes represent the 75th and 25th percentiles; the line within the box represents the median value; the whiskers above and below the boxes represent the 90th and 10th percentiles; the point within the box represents the mean value. Values adjacent to each confidence interval indicate number of measurement sites. The *n*

represents independent sites; more detail on this is presented in Section 2.2.
The effect values of SIA on the hazy days were significantly higher than those on
non-hazy days for all three periods (I, II, and III) (Fig. 5), indicating the persistent
prevalence of the SIA pollution problem over the past two decades. Considering
changes in concentrations, $SO_4^{2-}$ showed a downward trend from Period I to Period III
on the non-hazy days and hazy day, decreasing by 38.6% and 48.3%, respectively (Fig.
2b). These results reflect the effectiveness of the $SO_2$ pollution control policies (Ronald
et al., 2017). In contrast, there were no significant downward trends in concentrations
of $NO_3^-$ and $NH_4^+$ on either hazy or non-hazy days (Fig. 2c, d), but the mean $NO_3^-$
concentration in Period III decreased by 10.5% compared with that in Period II,
especially on hazy days (-16.8%). These results could be partly supported by decreased
$NO_x$ emissions and tropospheric $NO_2$ vertical column densities between 2011 and 2019
in China owing to effective $NO_x$ control policies (Zheng et al., 2018; Fan et al., 2021).
The lack of significantly downward trends in $NH_4^+$ concentrations is due to the fact that
the total $NH_3$ emissions in China changed little and remained at high levels between
2000 and 2018, i.e., slightly decreased from 2000 (10.3 Tg) to 2012 (9.3 Tg) (Kang et
al., 2016) and then slightly increased between 2013 and 2018 (Liu et al., 2021). The
same trends are also found in Quzhou in China, which is a long-term in situ monitoring
site (in Quzhou County, North China Plain, operated by our group; the detailed
information on Quzhou can be found in Meng et al. (2022) and Feng et al. (2022))
during the period 2012-2020 from previous publications (Xu et al., 2016; Zhang et al.,
2021b, noted that data during 2017-2020 are unpublished before) (Fig. S8). Zhang et
al. (2020b) found that the clean air actions implemented in 2017 effectively reduced
wintertime concentrations of $PM_1$ (particulate matter with diameter ≤1 μm), $SO_4^{2-}$ and

NH$_4^+$ in Beijing compared with those in 2007, but had no apparent effect on NO$_3^-$. Li et al. (2021) also found that SO$_4^{2-}$ exhibited a significant decline, However, NO$_3^-$ did not evidently exhibit a decreasing trend in the BTH region.

Our findings are to some extent supported by the nationwide measurements. Annual mean SO$_2$ concentrations displayed a clear decreasing trend with a 53% reduction in 2019 relative to 2015 for the four megacity clusters of eastern China (Fig. 3b, e), whereas there were only slight reductions in annual mean NO$_2$ concentrations (Fig. 3c, f). In contrast, annual mean NH$_3$ concentrations showed an obvious increasing trend in in both northern and southern regions of China, and especially in the BTH region (Fig. S9).

Overall, the above analyses indicate that SO$_4^{2-}$ concentrations responded positively to air policy implementations at the national scale, but that reducing NO$_3^-$ and NH$_4^+$ remains a significant challenge. China has a history of around 10-20 years for SO$_2$ and NO$_x$ emission control and has advocated NH$_3$ controls despite to date no mandatory measures and binding targets having been set (Fig. S2). Nevertheless, PM$_{2.5}$ pollution, especially SIA such as NO$_3^-$ and NH$_4^+$, is currently a serious problem (Fig. 4 and 5a, b). Some studies have reported that PM$_{2.5}$ pollution can be effectively reduced if implementing synchronous NH$_3$ and NO$_x$/SO$_2$ controls (Liu et al., 2019b). Therefore, based on the above findings, we propose that NH$_3$ and NO$_x$/SO$_2$ emission mitigation should be simultaneously strengthened to mitigate haze pollution.

**Fig. 5.** Variations in PM$_{2.5}$ composition, NS, NSN, SOR, and NOR from non-hazy to hazy days in (a) Period I (2000–2012), (b) Period II (2013–2016), (c) Period III (2017–2019). NS is the slope of the regression equation between [NH$_4^+$] and [SO$_4^{2-}$], NSN is the slope of the regression equation between [NH$_4^+$] and [SO$_4^{2-}$ + NO$_3^-$], SOR is sulfur oxidation ratio, and NOR is nitrogen oxidation ratio. The variations are statistically significant if the confidence intervals of the effect size do not overlap with zero. Values adjacent to each confidence interval indicate number of measurement sites. The *n* represents independent sites; more detail on this is presented in Section 2.2.

**3.2. Sensitivities from model simulations**

To further examine the efficiencies of NH$_3$ and acid gas emission reductions on SIA and PM$_{2.5}$ mitigation, the decreases of mean SIA and PM$_{2.5}$ concentrations with and without additional 50% NH$_3$ reductions were simulated using the WRF/CMAQ model.

Fig. 6 and Fig. S10 shows that, compared to 2010, SIA and $PM_{2.5}$ concentrations in January in 2017 were significantly decrease in the BTH, YRD, SCB, and PRD megacity clusters, respectively, in the simulations without additional $NH_3$ emission reductions. Across the four megacity clusters, the reduction in SIA and $PM_{2.5}$ is largest in the SCB region from 2010 to 2017 and smallest in the PRD region.

When simulating the effects of an additional 50% $NH_3$ emissions reductions in January in each of the years 2010, 2014 and 2017, the SIA concentrations in the megacity clusters (i.e. BTH, YRD, SCB and PRD) decreased by $25.9 \pm 0.3\%$, $24.4 \pm 0.3\%$, and $22.9 \pm 0.3\%$, respectively (Fig. 6 , Fig. S11, and Table S6). The reductions of $PM_{2.5}$ in 2010, 2014 and 2017 were $9.7 \pm 0.1\%$, $9.0 \pm 0.1\%$, and $9.2 \pm 0.2\%$ in the megacity clusters, respectively (Figs. S10 and S12). Whilst these results confirm the effectiveness of $NH_3$ emission controls, it is important to note that the response of SIA concentrations is less sensitive to additional $NH_3$ emission controls along the timeline of the $SO_2$ and $NO_x$ anthropogenic emissions reductions associated with the series of clean air actions implemented by the Chinese government from 2010 to 2017 (Zheng et al., 2018). Given the feasibility and current upper bound of $NH_3$ emission reductions options in the near future (50%) (Liu et al., 2019b), further abatement of SIA concentrations merely by reducing $NH_3$ emissions is limited in China. In other words, the controls on acid gas emissions should continue to be strengthened beyond their current levels.

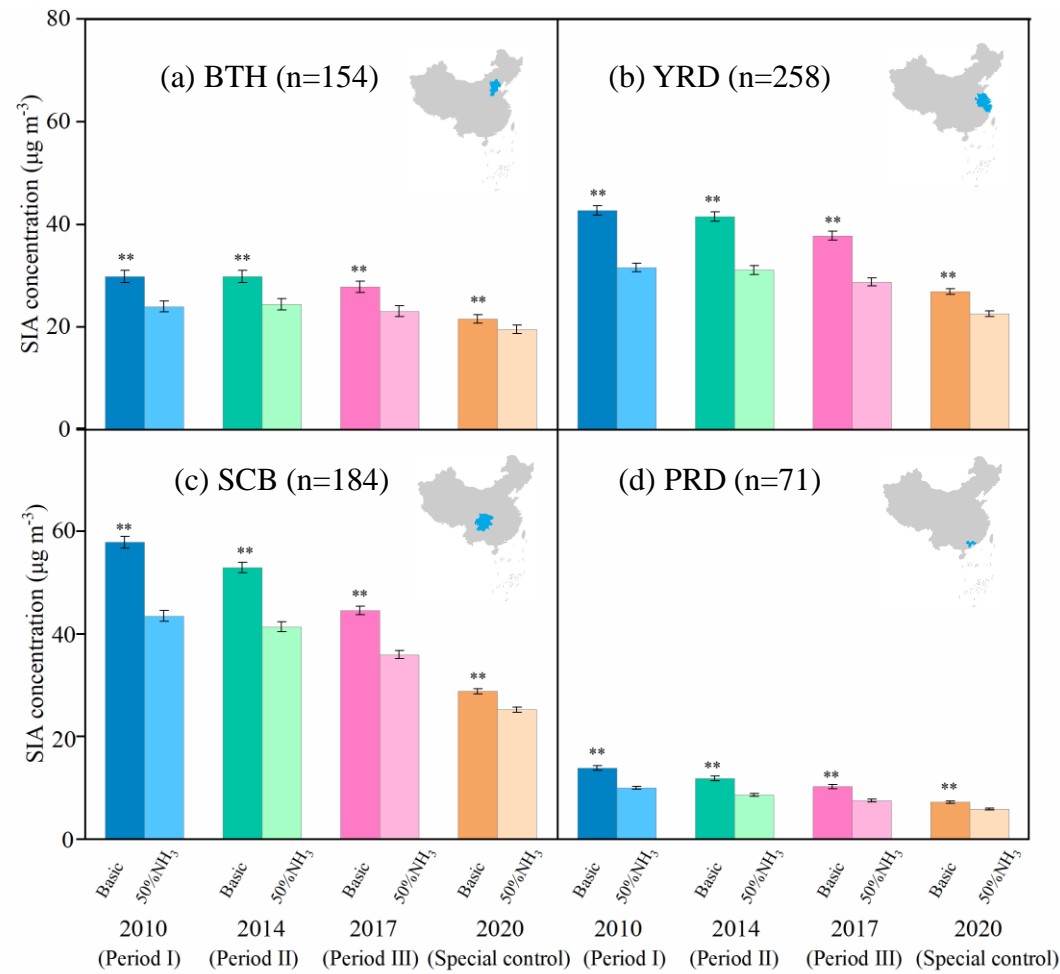

**Fig. 6.** Simulated SIA concentrations (in μg m$^{-3}$) without (basic) and with 50% ammonia (NH$_3$) emissions reductions in January for the years 2010, 2014, 2017 and 2020 in four megacity clusters (BTH: Beijing-Tianjin-Hebei, YRD: Yangtze River Delta, SCB: Sichuan Basin, PRD: Pearl River Delta). Inset maps indicate the location of each region. $^{**}$ denotes significant difference without and with 50% ammonia emission reductions ($P$ <0.05). $n$ is the number of calculated samples by grid extraction. Error bars are standard errors of means. (Period I (2000–2012), Period II (2013–2016), and Period III (2017–2019); Special control is the restrictions in economic activities and associated emissions during the COVID-19 lockdown period in 2020).

To further verify the above findings, we used the reductions of emissions of acid gases (46% and 23% for NO$_x$ and SO$_2$, respectively, in the whole China) during the

COVID-lockdown period as a further scenario (Huang et al., 2021). The model
simulations suggest that the effectiveness of reductions in SIA and $PM_{2.5}$ concentrations
by a 50% $NH_3$ emission reduction further declined in 2020 ($15 \pm 0.2\%$ for SIA, and
$5.1 \pm 0.2\%$ for $PM_{2.5}$), but the resulting concentrations of them were lower ($20.8 \pm 0.3\%$
for SIA, and $15.6 \pm 0.3\%$ for $PM_{2.5}$) when compared with that in 2017 under the same
scenario of an additional 50% $NH_3$ emissions reduction (and constant meteorological
conditions) (Fig. 6 and Table S6), highlighting the importance of concurrently $NH_3$
mitigation when acid gas emissions are strengthened. To confirm the importance of acid
gas emissions, another sensitivity simulation was conducted for 2017, in which the acid
gas ($NO_x$ and $SO_2$) emissions were reduced by 50% (Fig. 7). We found that reductions
in SIA concentrations are $13.4 \pm 0.5\%$ greater for the 50% reductions in $SO_2$ and $NO_x$
emissions than for the 50% reductions in $NH_3$ emissions. These results indicate that to
substantially reduce SIA pollution it remains imperative to strengthen emission controls
on $NO_x$ and $SO_2$ even when a 50% reduction in $NH_3$ emission is targeted and achieved.

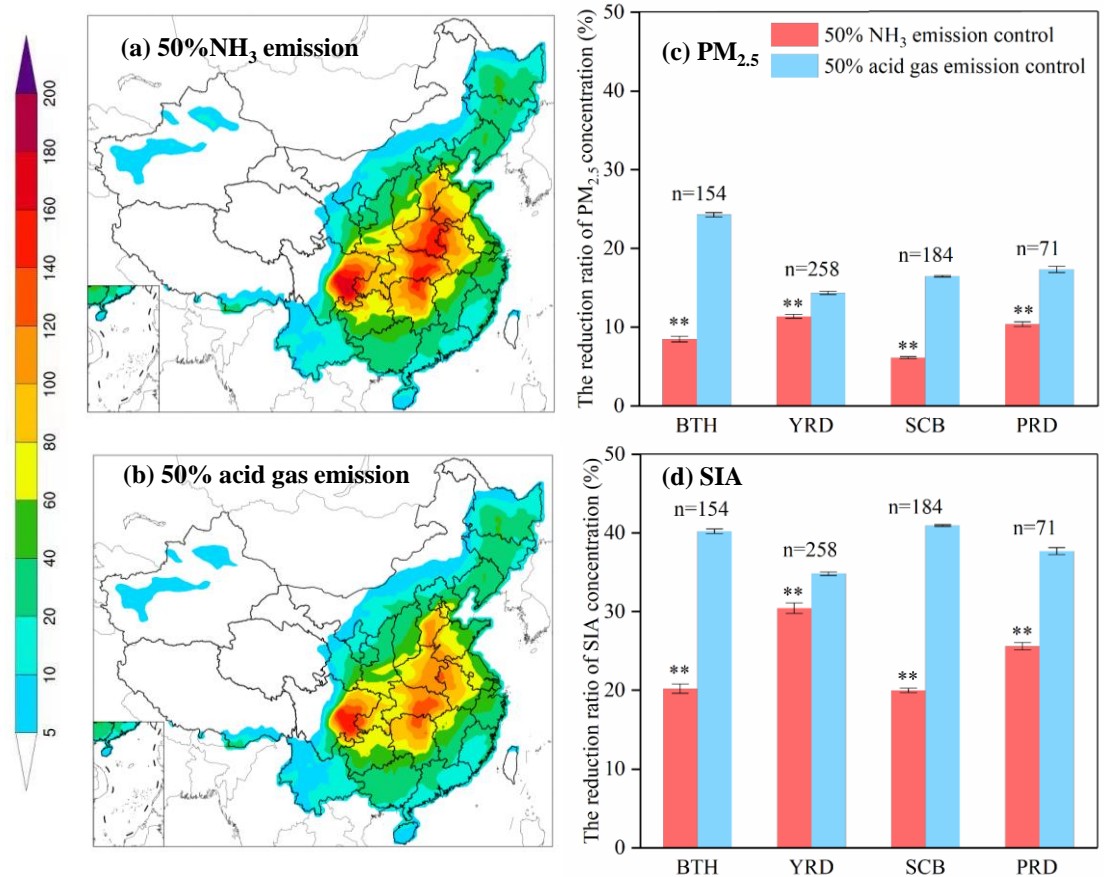


**Fig. 7.** Left: the spatial distributions of simulated $PM_{2.5}$ concentrations (in μg m$^{-3}$) in
January 2017 with (a) 50% reductions in ammonia ($NH_3$) emissions and (b) 50%
reductions in acid gas ($NO_x$ and $SO_2$) emissions. Right: the % decreases in $PM_{2.5}$ (c)
and SIA (d) concentrations for the simulations with compared to without the $NH_3$ and
acid gas emissions reductions in four megacity clusters (BTH: Beijing-Tianjin-Hebei,
YRD: Yangtze River Delta, SCB: Sichuan Basin, PRD: Pearl River Delta). ** denotes
significant differences without and with 50% ammonia emission reductions ($P < 0.05$).
$n$ is the number of calculated samples by grid extraction. Error bars are standard errors
of means.
**3.3. Uncertainty analysis and limitations**
Some limitations should be noted in interpreting the results of the present study: this
study examined period-to-period changes in $PM_{2.5}$ chemical components based on a

meta-analysis and the efficiencies of $NH_3$ and acid gas emission reductions on $PM_{2.5}$ mitigation. Some uncertainties may still exist in meta-analysis of nationwide measurements owing to differences in monitoring, sample handling and analysis methods as well as lack of long-term continuous monitoring sites (Fig. 2). For example, the measurements of $PM_{2.5}$ were mainly taken using the TEOM method, which is associated with under-reading of PM due to some nitrate volatilization at its operational temperature. To test whether the use of data during 2000–2019 could bias annual trends of $PM_{2.5}$ and chemical components, we summarize measurements of $PM_{2.5}$ at a long-term monitoring site (Quzhou County) during the period 2012-2020. The $PM_{2.5}$ and $SO_4^{2-}$ show the decreasing trend. The concentration of $NO_3^-$ and $NH_4^+$ do not show significant change (Fig. S8). The results are consistent with the trend for the whole of China obtained from the meta-analysis. Considering the uncertainty of $PM_{2.5}$ and its major components between different seasons (winter, summer, etc) and site type (urban, suburban or rural). We have analyzed historic trend in the different season and sites (Figs. S13-S20). We found that concentrations of $PM_{2.5}$ and its major chemical components ($SO_4^{2-}$, $NO_3^-$, and $NH_4^+$) were significantly higher in fall and winter than in spring and summer (Fig. S13). Only the winter season showed significant change trend in the three periods (Figs. S14-S17). The analyses also confirmed that pollution days predominated in winter. We also found that concentrations of $PM_{2.5}$ and its major chemical components were higher at urban than rural sites (Fig. S18). Spatially, the trends of $PM_{2.5}$ and its major components are similar across the whole of China (both of urban and rural) (Fig. S19). Rural areas show the same change trend in hazy days compared with whole of China (Fig. S20).

WRF-CMAQ model performance also has some uncertainty. We performed the validations of WRF and CMAQ models. The simulations of temperature at 2 m above

ground (T2), wind speed (WS), and relative humidity (RH) versus observed values at 400 monitoring sites in China are shown in Fig. S7. The meteorological measurements were obtained from the National Climate Data Center (NCDC) (ftp://ftp.ncdc.noaa.gov/pub/data/noaa/). The comparisons showed that the model performed well at predicting meteorological parameters with $R$ values of 0.94, 0.64 and 0.82 for T2, WS and RH, respectively. However, the WS was overestimated (22.3% NMB) in most regions of China, which is also reported in previous studies (Gao et al., 2016; Chen et al., 2019). This may be related to the underlying surface parameters set in the WRF model configurations.

In addition, the simulations of $PM_{2.5}$ and associated chemical components by the CMAQ model have potential biases in the spatial pattern, although the CMAQ model has been extensively used in air quality studies (Backes et al., 2016; Zhang et al., 2019) and the validity of the chemical regime in the CMAQ model had been confirmed by our previous studies (Zhang et al., 2021a; Wang et al., 2020a, 2021a). Since nationwide measurements of $PM_{2.5}$ and associated chemical components are lacking in 2010 in China, we undertook our own validation of $PM_{2.5}$ and its components (such as $SO_4^{2-}$, $NO_3^-$, and $NH_4^+$) using a multi-observation dataset that includes those monitoring data and satellite observations at a regional scale that were available.

First, the simulated monthly mean $PM_{2.5}$ concentration in January 2010 was compared with corresponding data obtained from the ChinaHighAirPollutants (CHAP, https://weijing-rs.github.io/product.html) database. The satellite historical $PM_{2.5}$ predictions are reliable (average $R^2 = 0.80$ and RMSE = 11.26 μg m$^{-3}$) using cross validation against the in-situ surface observations on a monthly basis (Wei et al., 2020, 2021). The model well captured the spatial distributions of $PM_{2.5}$ concentrations in our studied regions of BTH, YRD, PRD, and SCB (Fig. S3a), with correlation coefficient

(*R*) between simulated and satellite observed PM$_{2.5}$ concentrations of 0.96, 0.80, 0.60,
and 0.85 for BTH, YRD, PRD, and SCB, respectively.
Second, we also collected ground-based observations from previous publications
(Xiao et al., 2020, 2021; Geng et al., 2019; Xue et al., 2019) to validate the modeling
concentrations of SO$_4^{2-}$, NO$_3^-$, and NH$_4^+$. Detailed information about the monitoring
sites is presented in Table S5. The distributions of the simulated monthly mean
concentrations of SO$_4^{2-}$, NO$_3^-$, and NH$_4^+$ in January 2010 over China is compared with
collected surface measurements are shown in Fig. S4a, b, and c, respectively, with their
linear regression analysis presented in Fig. S4d. The model showed underestimation in
simulating SO$_4^{2-}$ and NO$_3^-$ in the BTH region, which might be caused by the uncertainty
in the emission inventory. The lack of heterogeneous pathways for SO$_4^{2-}$ formation in
the CMAQ model might also be an important reason for the negative bias between
simulations and measurements (Yu et al., 2005; Cheng et al., 2016). The model
overestimated NO$_3^-$ concentration in the SCB region, but can capture the spatial
distribution of NO$_3^-$ in other regions. The overestimation of NO$_3^-$ has been a common
problem in regional chemical transport models such as CMAQ, GEOS-CHEM and
CAMx (Yu et al., 2005; Fountoukis et al., 2011; Zhang et al., 2012; Wang et al., 2013),
due to the difficulties in correctly capturing the gas and aerosol-phase nitrate
partitioning (Yu et al., 2005). The modeling of NH$_4^+$ concentrations show good
agreement with the observed values. Generally, the evaluation results indicate that the
model reasonably predicted concentrations of SO$_4^{2-}$, NO$_3^-$, and NH$_4^+$ in PM$_{2.5}$.
Third, we performed a comparison of the time-series of the observed and simulated
hourly PM$_{2.5}$ and its precursors (SO$_2$ and NO$_2$) during January 2010. The model well
captures the temporal variations of the PM$_{2.5}$ in Beijing, with an NMB value of 0.05 μg
m$^{-3}$, NME of 28%, and *R* of 0.92 (Fig. 5a). The predicted daily concentrations of NO$_2$

and $SO_2$ during January 2010 also show good agreement with the ground measurements in Beijing, with NMB and $R$ values of 0.12 µg m$^{-3}$ and 0.89 for $NO_2$, and -0.04, 0.95 for $SO_2$, respectively (Fig. 5b). The variations of daily $PM_{2.5}$ concentrations between simulation and observation at 4 monitoring sites (Shangdianzi, Chengdu, Institute of Atmospheric Physics, Chinese Academy of Sciences (IAP-CAS), and Tianjin) from 14 to 30 January 2010 also matched well, with NMB values ranging from -0.05 to 0.12 µg m$^{-3}$, and $R$ values exceeding 0.89 (Fig. S5c).

We also compared the simulated and observed concentrations of $PM_{2.5}$, $NO_2$, and $SO_2$ in China in pre-COVID period (1–26 January 2020) and during the COVID-lockdown period (27 January–26 February) with actual meteorological conditions. As shown in Fig. S6, both the simulations and observations suggested that the $PM_{2.5}$ and $NO_2$ concentrations substantially decreased during the COVID-lockdown, mainly due to the sharp reduction in vehicle emissions (Huang et al., 2021; Wang et al., 2021b). For $SO_2$, the concentrations decreased very little and even increased at some monitoring sites. The model underestimated the concentrations of $PM_{2.5}$, $NO_2$, and $SO_2$, with NMB values of -21.4%, -22.1%, and -9.6%, respectively. We also newly evaluated the model performance in actual meteorological conditions for $PM_{2.5}$ concentrations in January 2014 and 2017, respectively. As shown in the Figure S21, the model well captured the spatial distribution of $PM_{2.5}$ concentration in China with MB (NMB) values of 23.2 µg m$^{-3}$ (15.4%) and 26.8 µg m$^{-3}$ (-26.7%) for 2014 and 2017, respectively. The simulated $PM_{2.5}$ concentrations compared well against the observations, with R values of 0.82 and 0.65, respectively

**3.4. Implication and outlook**

Improving air quality is a significant challenge for China and the world. A key target in China is for all cities to attain annual mean $PM_{2.5}$ concentrations of 35 μg m$^{-3}$ or below by 2035 (Xing et al., 2021). However, this study has shown that 74% of 1498 nationwide measurement sites have exceeded this limit value in recent years (averaged across 2015-2019). Our results indicated that acid gas emissions still need to be a focus of control measures, alongside reductions in $NH_3$ emissions, in order to reduce SIA (or $PM_{2.5}$) formation. Model simulations for the month of January underpin the finding that the relative effectiveness of $NH_3$ emission control decreased over the period from 2010 to 2017. However, simulating the substantial emission reductions in acid gases due to the lockdown during the COVID-19 pandemic, with fossil fuel-related emissions reduced to unprecedented levels, indicated the importance of ammonia emission abatement for $PM_{2.5}$ air quality improvements when $SO_2$ and $NO_x$ emissions have already reached comparatively low levels. Therefore, a strategic and integrated approach to simultaneously undertaking acid gas emissions and $NH_3$ mitigation is essential to substantially reduce $PM_{2.5}$ concentrations. However, the mitigation of acid gas and $NH_3$ emissions pose different challenges due to different sources they originate from.

The implementation of further reduction of acid gas emissions is challenging. The prevention and control of air pollution in China originally focused on the control of acid gas emissions (Fig. S2). The controls have developed from desulfurization and denitrification technologies in the early stages to advanced end-of-pipe control technologies. By 2018, over 90% of coal-fired power plants had installed end-of-pipe control technologies (CEC, 2020). The potential for further reductions in acid gas emissions by end-of-pipe technology might therefore be limited. Instead, addressing

total energy consumption and the promotion of a transition to clean energy through a de-carbonization of energy production is expected to be an inevitable requirement for further reducing $PM_{2.5}$ concentrations (Xing et al., 2021). In the context of improving air quality and mitigating climate change, China is adopting a portfolio of low-carbon policies to meet its Nationally Determined Contribution pledged in the Paris Agreement. Studies show that if energy structure adjusts and energy conservation measures are implemented, $SO_2$ and $NO_x$ will be further reduced by 34% and 25% in Co-Benefit Energy scenario compared to the Nationally Determined Contribution scenario in 2035 (Xing et al., 2021). Although it has been reported that excessive acid gas emission controls may increase the oxidizing capacity of the atmosphere and increase other pollution, $PM_{2.5}$ concentrations have consistently decreased with previous acid gas control (Huang et al., 2021). In addition, under the influence of low-carbon policies, other pollutant emissions will also be controlled. Opportunities and challenges coexist in the control of acid gas emissions.

In contrast to acid gas emissions, $NH_3$ emissions predominantly come from agricultural sources. Although the Chinese government has recognized the importance of $NH_3$ emissions controls in curbing $PM_{2.5}$ pollution, $NH_3$ emissions reductions have only been proposed recently as a strategic option and no specific nationwide targets have yet been implemented (CSC, 2018b). The efficient implementation of $NH_3$ reduction options is a major challenge because $NH_3$ emissions are closely related to food production, and smallholder farming is still the dominant form of agricultural production in China. The implementation of $NH_3$ emissions reduction technologies is subject to investment in technology, knowledge and infrastructure, and most farmers are unwilling or economically unable to undertake additional expenditures that cannot generate financial returns (Gu et al., 2011; Wu et al., 2018b). Therefore, economically

feasible options for $NH_3$ emission controls need to be developed and implemented
nationwide.
We propose the following three requirements that need to be met to achieve
effective reductions of SIA concentrations and hence of $PM_{2.5}$ concentrations in China.
First, binding targets to reduce both $NH_3$ and acid gas emissions should be set. The
targets should be designed to meet the $PM_{2.5}$ standard, and $NH_3$ concentrations should
be incorporated into the monitoring system as a government assessment indicator. In
this study, we find large differences in $PM_{2.5}$ concentration reductions from $NH_3$
emissions reduction in the four megacity regions investigated. At a local scale (i.e., city
or county), the limiting factors may vary within a region (Wang et al., 2011). Thus,
local-specific environmental targets should be considered in policy-making.
Second, further strengthening of the controls on acid gas emissions are still needed,
especially under the influence of low-carbon policies, to promote emission reductions
and the adjustment of energy structures and conservation. Ultra-low emissions should
be requirements in the whole production process, including point source emissions,
diffuse source emissions, and clean transportation (Xing et al., 2021; Wang et al.,
2021a). The assessment of the impact of ultra-low emissions is provided in Table S7.
In terms of energy structure, it is a requirement to eliminate outdated production
capacity and promote low-carbon new energy generation technologies.
Third, a requirement to promote feasible $NH_3$ reduction options throughout the
whole food production chain, for both crop and animal production. Options include the
following. 1) Reduction of nitrogen input at source achieved, for example, through
balanced fertilization based on crop needs instead of over-fertilization, and promotion
of low-protein feed in animal breeding. 2) Mitigation of $NH_3$ emissions in food
production via, for example, improved fertilization techniques (such as enhanced-

efficiency fertilizer (urease inhibitor products), fertilizer deep application, fertilization-irrigation technologies (Zhan et al., 2021), and coverage of solid and slurry manure. 3) Encouragement for the recycling of manure back to croplands, and reduction in manure discarding and long-distance transportation of manure fertilizer. Options for $NH_3$ emissions control are provided in Table S4. Although the focus here has been on methods to mitigate $NH_3$ emissions, it is of course critical simultaneously to minimize N losses in other chemical forms such as nitrous oxide gas emissions and aqueous nitrate leaching (Shang et al., 2019; Wang et al., 2020b).

**4. Conclusions**

The present study developed an integrated assessment framework using meta-analysis of published literature results, analysis of national monitoring data, and chemical transport modelling to provide insight into the effectiveness of SIA precursor emissions controls in mitigating poor $PM_{2.5}$ air quality in China. We found that $PM_{2.5}$ concentration significantly decreased in 2000-2019 due to acid gas control policies, but $PM_{2.5}$ pollution still severe. Compared with other components, this difference was more significant higher (average increase 98%) for secondary inorganic ions (i.e., $SO_4^{2-}$, $NO_3^-$, and $NH_4^+$) on hazy days than on-hazy days. This is mainly caused by the persistent SIA pollution during the same period. with sulfate concentrations significantly decreased and no significant changes observed for nitrate and ammonium concentrations. The reductions of SIA concentrations in January in megacity clusters of eastern China by additional 50% $NH_3$ emission controls decreased from $25.9 \pm 0.3\%$ in 2010 to $22.9 \pm 0.3\%$ in 2017, and to $15 \pm 0.2\%$ in the COVID lockdown in 2020 for simulations representing reduced acid gas emissions to unprecedented levels, but the SIA concentrations decreased by $20.8 \pm 0.3\%$ in 2020 compared with that in 2017 under the same scenario of an additional 50% $NH_3$ emissions reduction. In addition, the reduction

of SIA concentration in 2017 was $13.4 \pm 0.5\%$ greater for 50% acid gas ($SO_2$ and $NO_x$) reductions than for the $NH_3$ emissions reduction. These results indicate that acid gas emissions need to be further controlled concertedly with $NH_3$ reductions to substantially reduce $PM_{2.5}$ pollution in China.

Overall, this study provides new insight into the responses of SIA concentrations in China to past air pollution control policies and the potential balance of benefits in including $NH_3$ emissions reductions with acid gas emissions controls to curb SIA pollution. The outcomes from this study may also help other countries seeking feasible strategies to mitigate $PM_{2.5}$ pollution.

**Data availability**

All data in this study are available from the from the corresponding authors (Wen Xu, wenxu@cau.edu.cn; Shaocai Yu, shaocaiyu@zju.edu.cn) upon request.

**Author contributions**

W.X., S.Y., and F.Z. designed the study. F.M., Y.Z., W.X., and J.K. performed the research. F.M., Y.Z., W.X., and J.K. analyzed the data and interpreted the results. Y.Z. conducted the model simulations. L.L. provided satellite-derived surface $NH_3$ concentration. F.M., W.X., Y.Z., and M.R.H. wrote the paper, S. R., M.W., K.W., J.K., Y.Z., Y.H., P.L., J.W., Z.C., X.L., M.R.H., S.Y. and F.Z. contributed to the discussion and revision of the paper.

**Declaration of Competing Interest**

The authors declare that they have no known competing financial interests or personal relationships that could have appeared to influence the work reported in this paper.

**Acknowledgments**

This study was supported by National Natural Science Foundation of China (42175137, 21577126 and 41561144004), China Scholarship Council (201913043), the National Key Research and Development Program of China (2021YFD1700900), the Department of Science and Technology of China (2016YFC0202702, 2018YFC0213506 and 2018YFC0213503), National Research Program for Key Issues in Air Pollution Control in China (DQGG0107), and the High-level Team Project of China Agricultural University. SR's contribution was supported by the Natural Environment Research Council award number NE/R000131/1 as part of the SUNRISE program delivering National Capability. We would like to thank Prof. Yuepeng Pan and Dr. Yangyang Zhang for their help on validating the modeling results.

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
