# Peer review of "Trends in secondary inorganic aerosol pollution in China and its responses to"

_Atmospheric Chemistry and Physics, 2021_

## Referee Comment (RC2)

Interactive comment on "Trends in secondary inorganic aerosol pollution in China and its responses to emission controls of precursors in wintertime"

The authors analyzed the trends in PM2.5 and SIA observations collected from literature and observations from national monitoring network in China. They also conducted some model simulations to calculate the sensitivities of SIA to its precuros emissions changes and compared the efficiencies of reducing different precursors emissions in mitigating SIA pollution. Based on these simulated efficiencies, they proposed some requirements to further reduce SIA pollution. This topic is interesting and important (but not new) and within the scope of ACP. However, the results of this work is not reliable because the trend analysis is problematic and the model simulations has not been evaluated using observations. Also, some important questions about the drivers of the SIA trends are not addressed and an in-depth anlaysis exploring the drivers of the trends is needed. I think this work needs a thorough revision to address the questions and comments below. So I would suggest rejection and resubmission after addressing those issues.

**Major comments:**

- 1. When analyzing the trends of PM2.5 and SIA components using measurments collected from literature, the number of sites differ by a factor of four through the three periods. This make me concern about the reliability of the trends reported in this study. I think the authors should use the same sites for trend analysis to keep consistency.
- 2. Also, when dicussing the trends based on meta-analysis, the authors left a lot of key questions unexplained. For example, why did the sulfate concentrations during hazy days increased from period I to period II while a series of SO2 control policies has been implemented? Why did the nitrate concentrations not respond to the air pollution control policies from 2000 to 2019? In addition, an in-depth analysis about the drivers of the trends is lacking. The current manuscript just simply relates the trends with air pollution control policies and did not provide any quantitative analysis on the contributions from emission changes and meteorogolical impacts given that meteorological impacts can be much larger than the impacts from emission reductions (Sulaymon et al., 2021).
- 3. For all the simulations in this study, the authors did not provide any evaluation against measurements. Especially for the base simulations in sensivitiy calculation, you need to first evaluate your simulated chemical regime in the SIA formation before you conducting the NH3/NOx/SO2 emission reduction experiments and calculcating the sensitivities of SIA (PM2.5) formation to precurosors emission changes. So you need to first evaluate your simulated sulfate, nitrate, ammonium, SO2, NO2, and NH3 using measurements.
- 4. The authors examed the trends of SIA and PM2.5 based on observations collected from literature and explored the efficiency of NH3 and acidic gases emission reduction using model simulations. However, they didn't build any connection between these two parts. They actually can use the observations to evaluate the simulated chemical regime before calculating the emission reduction efficiency. Or they can use model simulations to explore the drivers of the trends in the observed SIA and PM2.5 concentrations through the three periods.

- 5. Also, while the meta-analysis shows that the nitrate concentrations do not significantly respond to air pollution control policies, the SIA sensitivity simulations show large decreases when reducing acidic gases emissions. Here I think the authors need to check whether the simulated nitrate concentrations decrease or not when reducing NOx emissions and see if they are consistent with the observed nitrate concentration changes.
- 6. Some of the references are not appropriate and do support their text.

**Specific comments:**

- 1. Fig. 2: what does n represent? number of sites? The number of sites for the three periods differ by a factor of four (e.g. 93 vs 25 in Fig. 2 (a))? I think you need to use the same sites through the three periods to analyze the trends.
- 2. Fig. 2: add "observed" or "measured" before "concentrations" in line 277 to make it clear that these data are measurements, not simulations. Same for Fig.3.
- 3. Also in Fig. 3, when you analyze the trends for each region, you need to use the same sites through the five years.
- 4. Fig.3: why do you skip the years before 2015 given that you analyze trends from 2000 to 2019 in Fig. 2 ?
- 5. Line 264-266: How significant is the decreasing trend of 19.9%? Also, both the PM2.5 concentrations during hazy and non-hazy days increased from period I to period II, which contradicts with line 270-271 and Fig. S2. What caused the increases in PM2.5 concentrations between period I and period II?
- 6. Line 422: did you reduce NOx and SO2 emissions by 50% simultaneously?
- 7. Line 305: what do you mean by 46 groups of data? do you mean data from 46 sites, including both measurement during hazy and non-hazy periods?
- 8. Fig.4 (A): what are the numbers on the right of the error bars? The number of sites?
- 9. Line 306-313: what's the cause for the changes? meteorology (e.g. wind, precipitation), emissions or chemistry? I think here you need to consider the weather condition when you classify hazy or non-hazy days.
- 10. Line 308-313 contradict with line 313-317: while your data shows no significant difference in the SIA portion (36-40%) between hazy and non-hazy days, you conclude SIA is the dominant role in haze pollution? In addition, in Fig 4. (B), 'other' plus OC is greater than 50%. What is 'other' in Fig. 4 (B)?
- 11. Fig. 5: again, if you want to compare the metrics from different periods, you need to use measurements from the same sites to keep consistency. Here, the number of sites for nitrate

differ by a factor of 4. Also, the range of the x axis should be the same for these three plots for comparison.

- 12. Line 332: what do you mean by "effect values"
- 13. Line 335-338: 19.9% decrease (in average or in the median value?) from which period to which period? 49.6% decrease from which period to which period? Did you check the meteorology change (e.g. wind, precipitation, etc.) during the three periods? How can you make sure it's the SO2 control policy not the meteorology change that caused the decrease in sulfate? Also, how do you explain the increase of sulfate during hazy days from period I to period II while you claim the SO2 control policies were effective?
- 14. Line 338-341 and line 350-351: So here do you mean that the NOx control policies since 2011 were not effective? If this is the case, how do you explain the difference between your conclusion and Fan et al. (2021), which reports decreasing trends in NO2 observations in China from 2011 to 2019 owing to effective NOx control policies?
- 15. Line 341-343: Kang et al. (2016) (Figure 1) shows a decreasing trend in Chinese total NH3 emissions from 2000 to 2012 and doesn't show any further trends after 2012. How can this explain the "the lack of downward trends in NH4+" in your Fig. 2d?
- 16. Line 344-347: In Zhang et al. (2020) (the reference between line 818-821), I didn't find any data supporting your sentences here.
- 17. Line 348-353: again, please make sure you are comparing the same sites for each region through these years.
- 18. Line 354-356: so what do you think is the reason that nitrate concentrations did not significantly respond to air pollution mitigation policies? Is it because NOx emissions did not really decrease? Or is it because the chemistry regime was actually NH3-limited so that reducing NOx emission is not effective in reduing nitrate?
- 19. Fig. 4 (A) and Fig. (5): how did you calculate the "variation"? Is it acturally the ratio of the difference between concentrations during hazy and non-hazy days to the concentrations during non-hazy days?
- 20. Line 358-363: Fig. 4 (B) (b) shows that ammonium and nitrate only account for 20-23% of total PM2.5 during both hazy and non-hazy days. And only 3% difference is found in their contribution (%) between hazy and non-hazy days. This seems to not support your sentences that nitrate and ammonium are currently a serious problem given that "other" plus OC contribute more than 50% of total PM2.5. Also, line 360, where is the sub figure (d) in Fig. 4 and Fig. 5?
- 21. Figure S4: there seems to be a large bias in your simulated wind speed? Can you calculate the normalized mean bias for the comparisons?

- 22. Fig. 6: Your simulated SIA conncentrations over BTH are lower than those over YRD from 2010 to 2020. Have you evaluated your simulations (sulfate, nitrate, ammonium, total PM2.5) using measurements?
- 23. Line 376-378: I don't see any significant decreases in simulated SIA concentrations over BTH from 2010 to 2017 in your simulations without NH3 emissions reductions (Fig. 6). Also, did you evaluate your simulated trends of SIA and PM2.5 using measurements?
- 24. Line 379-380: why? Is it because that Sichuan has larger air polluants emission reductions than PRD? Did you check the meteorology change? Most importantly, did you evaluate this using measuremnets?
- 25. Line 384-385: I think the percentage reductions in simulated PM2.5 is much smaller than those in SIA.
- 26. Line 446-459: The PM2.5 dataset from STET model are not real "observations". In addition, your PM2.5 simulation show significant bias compared to the STET data. You need to evaluate your simulated SIA components and SO2/NO2/NH3 using real observations and see if your simulated chemical regime is close to the true state or not. You already collected so many observations of SIA components, which can be used to evaluate your SIA simulations. Also, SO2/NO2/NH3 observations are available from multiple satellite insturments.
- 27. Line 547-549: It seems that Fig. 2 (a) only show small decreases of PM2.5 from 2000 to 2019 during non-hazy days, and no significant dereases were found during hazy days. Most importantly, the trends here are not realiable because the number of sites in your trend analysis differ by a factor of four.
- 28. Line 551-559: again, without any evalution based on measurements of nitrate, sulfate, ammonium, NH3, SO2 and NO2, your sensitivity cacluations here are not reliable.

**Reference:**

- Fan, C., Li, Z., Li, Y., Dong, J., van der A, R., and de Leeuw, G.: Variability of NO2 concentrations over China and effect on air quality derived from satellite and groundbased observations, Atmos. Chem. Phys., 21, 7723–7748, https://doi.org/10.5194/acp-21-7723-2021, 2021
- Sulaymon, I. D., Zhang, Y., Hopke, P. K., Hu, J., Zhang, Y., Li, L., ... & Zhao, F. (2021). Persistent high PM2. 5 pollution driven by unfavorable meteorological conditions during the COVID-19 lockdown period in the Beijing-Tianjin-Hebei region, China. Environmental Research, 198, 111186.

---

## Author Comment (AC1)

We thank the reviewers for their comments, which have helped us substantially to improve our manuscript. Below, we explain how we incorporated the comments into the revised version. Our responses are given in blue below, and revisions to the manuscript are shown in track changes (with line number references).

**Reviewer#1**

1.The study examined annual trends in $PM_{2.5}$ chemical components based on a meta-analysis and the efficiencies of $NH_3$ and acid gas emission reductions on $PM_{2.5}$ mitigation. The authors also looked at hazy vs non-hazy days, yet the abstract doesn't mention them – could this be addressed?

**Response:** As suggested by the reviewer, in the revised paper we have added information about hazy days and non-hazy days to the Abstract as follows: "The concentration of $PM_{2.5}$ and its components were significantly higher (16%-195%) on hazy days than on non-hazy days. Compared with mean values of other components, this difference was more significant for the secondary inorganic ions $SO_4^{2-}$, $NO_3^-$, and $NH_4^+$ (average increase 98%)" (See track changes in Lines 40-44 in the revised manuscript).

2.The CMAQ model run undertakes a 50% reduction in $NH_3$ but only for January – very little comment is made of why this month was chosen and how this relates to an annual average. Comment on whether 50% reduction is a realistic target for the Chinese Government.

**Response:** The following text in the revised manuscript explains our choice of January in more detail (See track changes in Lines 246-250 in the revised manuscript): "January was selected as the typical simulation month because wintertime haze pollution

frequently occurs in this month (Wang et al., 2011; Liu et al., 2019b). The sensitivity scenarios of emissions in January can therefore help to identify the efficient option to control haze pollution."

Yes, a 50% reduction in $NH_3$ emissions is a realistic target for China. Zhang et al. (2020) found that the mitigation potential of $NH_3$ emissions from cropland production and livestock production in China can reach up to 52% and 58%, respectively. In addition, it is essential to jointly control agricultural $NH_3$ for China to achieve more stringent $PM_{2.5}$ goals in the future. This is echoed in the results of the project "National Research Program for Key Issues in Air Pollution Control", which reported that a 50% $NH_3$ emission reduction (e.g., from 1.6 to 0.81 Tg $yr^{-1}$) is necessary to achieve the proposed annual mean $PM_{2.5}$ target (35 $\mu g\ m^{-3}$) in the "2+26 cities" region of China.

To make this clearer, in the revised paper we now state that "The choice of 50% additional $NH_3$ emissions reduction is based on the feasibility and current upper bound of $NH_3$ emissions reduction expected to be realized in the near future (Liu et al., 2019b; Table S4). Zhang et al. (2020) found that the mitigation potential of $NH_3$ emissions from cropland production and livestock production in China can reach up to 52% and 58%, respectively." (See track changes in Lines 261-266 in the revised manuscript).

3.The authors spend a lot of time undertaking a meta analysis of the literature in order to put a database of secondary PM measurements together and this seems to have been done thoroughly, although I am not suitably familiar enough with the methods to comment further.

**Response:** In the revised paper, we have added the following brief introduction on Meta-analysis method in the Materials and methods: "Meta-analyses can be used to

quantify the differences in    concentrations of PM$_{2.5}$ and its secondary inorganic aerosol components (NH$_4^+$, NO$_3^-$, and SO$_4^{2-}$) between hazy    and non-hazy days and to identify the major pollutants on non-hazy days (Wang et al., 2019b); this provides evidence for effective options on control of precursor emissions (NH$_3$, NO$_2$, and SO$_2$) for reducing occurrences of hazy days." (See track change in Lines 148-153 in the revised manuscript).

4. I don't think the CMAQ model has been evaluated for Jan 2010 using measurements of PM or PM components, although there was some evaluation of met. parameters - temperature looked good RH and especially Wind Speed were quite poor (Fig s4) – note R was 0.5 on the wind speed graph but 0.64 in the text? There was a comparison between the CMAQ and STET model (defined as 'observations') but these were just two maps side by side. I'm not sure whether the STET model comparison is for the same period.

**Response**: We thank the reviewer for pointing out this mistake. In the revised paper, we have corrected the R (0.64) between the simulated and observed wind speed (Fig S7). We think that the corrected R value is an acceptable modelling result, as the overestimation of wind speed was a common problem in the WRF model, as widely reported in previous studies (Gao et al., 2016; Chen et al.,2019).    In the revised paper, we now include in Section 3.3 the following additional text on the validation of WRF model performances. (See track changes in lines 529-538 in the revised manuscript):

[revised manuscript text omitted]

**Figure S5**. Time series of the observed (red dots) and simulated (black line) (a) hourly concentrations of PM$_{2.5}$ and (b) daily concentrations of NO$_2$ and SO$_2$ in January 2010 in Beijing; (c) daily concentrations of PM$_{2.5}$ during 14-30 January 2010 at monitoring sites in Shangdianzi, Chengdu, Institute of Atmospheric Physics, Chinese Academy of Sciences (IAP-CAS) and Tianjin. The normalized mean bias (NMB) normalized mean error (NME), and correlation coefficient (*R*) are given in the plots.

5. No evaluation of CMAQ modelled components was made either, which makes one wonder whether it did predict well in Jan 2010. Without this the conclusions are weakened somewhat. I think to have more confidence in the results more should be

made of the evaluation against $PM_{2.5}$ and if possible PM components.

**Response**: We have provided full detail of our new model evaluation in response to comment #4 above. In brief again, for our revised paper we collected ground-based observations from the literature to verify the performance of the model of $PM_{2.5}$ and its chemical compositions in the following three ways:

First, the simulated monthly mean $PM_{2.5}$ concentration in January 2010 was compared with corresponding data from obtained from TAP database.

Second, the distribution of simulated monthly mean concentration of $SO_4^{2-}$, $NO_3^-$ and $NH_4^+$ in January 2010 over China compared with surface measurements are shown in Fig. S4a, b, and c, respectively, with their linear regression analysis presented in Fig. S4d.

Third, we performed a comparison of the time series of the observed and simulated hourly $PM_{2.5}$ and its precursors ($SO_2$ and $NO_2$) during January 2010.

The discussion of the results of these model validations are also presented in our response to comment #4 above and added to the revised paper.

6. It would have been useful for the authors to undertake a comparison of the CMAQ model predictions, associated with changing COVID emissions, and the actual measured changes.

**Response**: Thank you for this interesting suggestion. We have undertaken the suggestion of the reviewer in our revised paper. See the following additional text in track changes in lines 591-602 in the revised manuscript. "We also compared the simulated and observed concentrations of $PM_{2.5}$, $NO_2$, and $SO_2$ in China in pre-COVID period (1–26 January 2020) and during the COVID-lockdown period (27 January–26 February). As shown in Fig. S6, both the simulations and observations suggested that

the PM2.5 and NO₂ concentrations substantially decreased during the COVID-lockdown, mainly due to the sharp reduction in vehicle emissions (Huang et al., 2021; Wang et al., 2021b). For SO₂, the concentrations decreased very little and even increased at some monitoring sites. The model underestimated the concentrations of PM$_{2.5}$, NO$_2$, and SO$_2$, with NMB values of -21.4%, -22.1%, and -9.6%, respectively. This phenomenon is reasonable as the simulations for the two periods in 2020 used the meteorology for 2010 whereas measured changes are strongly influenced by the actual meteorological conditions."

[Figure]

**Figure S6.** Scatter plots of CMAQ simulations versus surface observations for PM$_{2.5}$, NO$_2$, and SO$_2$ concentrations before the COVID-lockdown (black dots) and during the COVID-lockdown period (red dots).

7. The measurements of PM$_{2.5}$ were taken using TEOM's although no mention was made of the associated problems under reading PM associated with nitrate and operational temperature, which common to these instruments. This is especially important since the paper focuses on SIA

**Response:** We agree that there may be systematic error using TEOM methodology. In the revised paper, we now state that "Some uncertainties may still exist in meta-analysis

of nationwide measurements owing to differences in monitoring, sample handling and analysis methods as well as lack of long-term continuous monitoring sites (Fig. 2). For example, the measurements of $PM_{2.5}$ were mainly taken using TEOM method, which is associated with under-reading of PM due to some nitrate volatilization at its operational temperature." (See track changes in lines 496-505 in the revised manuscript).

**Results**

8. As a general comment a lot of analysis has been made between Hazy and non-Hazy days, but the conclusions and abstract don't seem to reflect this.

**Response:** As suggested by the reviewer, we added the information about hazy and non-hazy days information to the Abstract: "The concentration of $PM_{2.5}$ and its component were significantly higher (16%-195%) on hazy days than on non-hazy days. Compared with mean values of other components, this difference was more significant for the secondary inorganic ions $SO_4^{2-}$, $NO_3^-$, and $NH_4^+$ (average increase 98%)". We also added the following information to the conclusions: "Compared with other components this difference was more significant (average increase 98%) for secondary inorganic ions (i.e., $SO_4^{2-}$, $NO_3^-$, and $NH_4^+$) on hazy days than on-hazy days" (See tack changes in lines 40-44 and lines 697-699 in the revised manuscripts).

9. For the trend analysis (fig 2) suggests a 19% reduction of PM2.5 between period 1 and 3 on non-hazy days although all of the box plots are for different numbers of sites and so it would be hard to say whether this is true? also the concentrations seemed to increase in period 2? Are these trends significant?

**Response:** Thank you for pointing this out. We now realize that the trend analysis in our study has some uncertainties. The historic trend analysis at the same sites were limited due to lack of long-term in situ measurements. In order to reduce the uncertainty of trend analysis, we have made some improvement in data analysis in the revised paper, as follows:

First, we re-filtered the data for meta-analysis and then made a three-period comparison using the measurements at sites that include both $PM_{2.5}$ and secondary inorganic ions ($SO_4^{2-}$, $NO_3^-$, and $NH_4^+$) (See tack changes in lines 298-304 in the revised manuscripts and updated Fig. 2).

Second, our statistical analysis on the concentrations of $PM_{2.5}$ and secondary inorganic ions for three periods now uses a non-parametric statistical method since concentrations were not normally distributed based on the Kruskal-Wallis test (Kruskal and Walls, 1952). For each species, the Kruskal-Wallis one-way analysis of variance (ANOVA) on ranks among three periods was performed with pairwise comparison using Dunn's method (Dunn, 1964). (See track changes in Lines 201-207 in the revised manuscript).

Third, to test whether the use of data during 2000-2019 could bias annual trends of $PM_{2.5}$ and chemical components, we summarize measurement of $PM_{2.5}$ at a long-term monitoring site (in Quzhou County, North China Plain, operated by our group) during the period 2012-2020 from previous publications (Xu et al., 2016; Zhang et al., 2021, noted that data during 2017-2020 are unpublished before) (Figure S8). The results are

consistent with trend in China from the meta-analysis (See track changes in lines 396-400 and lines 507-515 in the revised manuscript).

[Figure]

**Figure 2.** Comparisons of observed concentrations of (a) PM$_{2.5}$, (b) SO$_4^{2-}$, (c) NO$_3^-$, and (d) NH$_4^+$ between non-hazy and hazy days in Period I (2000–2012), Period II (2013–2016), and Period III (2017–2019). Bars with different letters denote significant differences among the three periods ($P$ <0.05) (upper and lowercase letters for non-hazy and hazy days, respectively). The upper and lower boundaries of the boxes represent the 75th and 25th percentiles; the line within the box represents the median value; the whiskers above and below the boxes represent the 90th and 10th percentiles; the point within the box represents the mean value. Comparison of the pollutants among the three-periods using Kruskal-Wallis and Dunn's test. The $n$ represents independent sites, more detail information on this is presented in Section 2.2.

[Figure]

**Figure S8**. Daily and monthly concentration of (a) PM$_{2.5}$, (b) SO$_4^{2-}$, (c)NO$_3^-$, and (d) NH$_4^+$ in Quzhou in China during 2002-2019.

10. Since the measurements are combined into periods the true trends are difficult to interpret. I think a description of a PM$_{2.5}$ timeseries for a site throughout the period

would be beneficial. With some comment on things like seasonality and reasons for the measurement trends. Most trends are ascribed to Government policy, although with the changes that have taken place in China, this may well be too simple.

**Response:** We thank the reviewer for their comments. To test whether the use of data during 2000-2019 could bias annual trends of $PM_{2.5}$ and chemical components, we summarize measurements of $PM_{2.5}$ at long-term monitoring site (in Quzhou County, North China Plain, operated by our group) during the period 2012-2020 from previous publications (Xu et al., 2016; Zhang et al., 2021, noted that data during 2017-2020 are unpublished before). The $PM_{2.5}$ and $SO_4^{2-}$ show the same decreasing trend. The concentrations of $NO_3^-$ and $NH_4^+$ do not show significant changes (Fig.S8). The results are consistent with the trend for whole of China obtained from the meta-analysis. (See track changes in lines 507-515 in the revised manuscript and Fig S8).

11. The authors mention the results in Fig 2a (page 11) and b,c,d, (page 16) which makes it hard for the reader. Consider revising the diagrams.

Response: Thanks for your suggestions. In the revised paper, we have added a more detail caption to Fig. 2: "Comparisons of observed concentrations of (a) $PM_{2.5}$, (b) $SO_4^{2-}$, (c) $NO_3^-$, and (d) $NH_4^+$ between non-hazy and hazy days in Period I (2000–2012), Period II (2013–2016), and Period III (2017–2019). Bars with different letters denote significant differences among the three periods ($P <0.05$) (upper and lowercase letters for non-hazy and hazy days, respectively). The upper and lower boundaries of the boxes represent the 75th and 25th percentiles; the line within the box represents the median value; the whiskers above and below the boxes represent the 90th and 10th percentiles; the point within the box represents the mean value. Comparison of the pollutants among

the three-periods using Kruskal-Wallis and Dunn's test. The *n* represents independent sites; more detail on this is presented in Section 2.2." (See track changes in lines 317-327 in the revised manuscript).

12. The authors spend quite a long time stating that $PM_{2.5}$ on hazy days is greater than on non-hazy days which seems fairly obvious given that the meta analysis chose data in this way.

**Response:** Our interest is in understanding which components within $PM_{2.5}$ are particularly elevated on hazy days relative to other components. This provides evidence for effective options on control of precursor emissions ($NH_3$, $NO_2$, and $SO_2$) for reducing occurrences of hazy days. As per our response to comment #3 above we now provide additional explanation of this aim in the Materials and methods section of the revised manuscript as follows. "Meta-analyses can be used to quantify the differences in concentrations of $PM_{2.5}$ and its secondary inorganic aerosol components ($NH_4^+$, $NO_3^-$, and $SO_4^{2-}$) between hazy and non-hazy days and to identify the major pollutants on non-hazy days (Wang et al., 2019b); this provides evidence for effective options on control of precursor emissions ($NH_3$, $NO_2$, and $SO_2$) for reducing occurrences of hazy days." Also, as per responses above, we have highlighted more the finding that the secondary inorganic ions (i.e., $SO_4^{2-}$, $NO_3^-$, and $NH_4^+$) were more elevated (higher on average by 98%) on hazy days than the elevation of other components. The meta-analysis approach can help us better understand the reason of $PM_{2.5}$ formation (See track change in Lines 42-44 and Lines 148-153 in the revised manuscript).

13. It says that SIA is a major influencing factor for haze pollution, yet in Fig 4 B (b) the proportion of total $PM_{2.5}$ is about the same as non Hazy day 40% vs 36%

respectively, suggesting that SIA goes up but so do other components of PM.

**Response:** Although the difference is not great (as the reviewer points out) it is nevertheless the case that the proportion of SIA components is higher on hazy days compared with non-hazy days. As we have noted in responses above, compared with other components the increase in concentrations was more significant (average increase of 98%) for the secondary inorganic ions $SO_4^{2-}$, $NO_3^-$, and $NH_4^+$ (see Figs 4A and 5).

14. There is very little mention of the other components of $PM_{2.5}$, OC, EC and the 'other' components, all of which are important – plus no model evaluation of these.

**Response:** Whilst OC and EC are important components of $PM_{2.5}$, their concentrations are not affected by $NO_x$, $SO_2$ or $NH_3$ emission reductions. Our research focus here is on the secondary inorganic aerosol pollution and therefore we pay less attention to the changes of OC and EC content. In response to other comments from this reviewer we have now undertaken extensive evaluation of the model performance for the SIA components, as described in detail above in response to comment #4. For one aspect of model evaluation the distribution of simulated monthly mean concentration of $SO_4^{2-}$, $NO_3^-$ and $NH_4^+$ in January 2010 over China was compared with surface measurements in Fig. S4a, b, and c, respectively, with their linear regression analysis showing in Fig. S4d. In a second evaluation, we compared the time series of the observed and simulated hourly $PM_{2.5}$ and its precursors ($SO_2$ and $NO_2$) during January 2010.

15. I hope these comments are useful

**Response:** We appreciate the reviewer for acknowledging the importance of our work. We also thank the reviewer for the constructive comments to improve our manuscript.

---

## Author Comment (AC2)

Response: Thanks for the referee's thoughtful and critical comments on our manuscript. Below we provide a point-by-point response to the reviewer' comments and how we have addressed them in the revised manuscript (in blue).

**Reviewer# 2**

Interactive comment on "Trends in secondary inorganic aerosol pollution in China and its responses to emission controls of precursors in wintertime"

The authors analyzed the trends in $PM_{2.5}$ and SIA observations collected from literature and observations from national monitoring network in China. They also conducted some model simulations to calculate the sensitivities of SIA to its precursors emissions changes and compared the efficiencies of reducing different precursors emissions in mitigating SIA pollution. Based on these simulated efficiencies, they proposed some requirements to further reduce SIA pollution. This topic is interesting and important (but not new) and within the scope of ACP. However, the result of this work is not reliable because the trend analysis is problematic and the model simulations has not been evaluated using observations. Also, some important questions about the drivers of the SIA trends are not addressed and an in-depth analysis exploring the drivers of the trends is needed. I think this work needs a thorough revision to address the questions and comments below. So, I would suggest rejection and resubmission after addressing those issues.

**Response:** We thank the reviewer for their constructive comments on our manuscript. The historic trend analysis at same sites were limited due to lack of long-term in situ measurements. In order to reduce the uncertainty of trend analysis, we have made improvements in data analysis in the revised paper via the following three approaches. First, we re-filtered the data for meta-analysis and then made a three-period

comparison using the measurements at sites that include both $PM_{2.5}$ and secondary inorganic ions ($SO_4^{2-}$, $NO_3^-$, and $NH_4^+$) (See tack changes in lines 298-304 in the revised manuscripts and updated Fig. 2). Second, our statistical analysis on the concentrations of $PM_{2.5}$ and secondary inorganic ions for three periods now uses a non-parametric statistical method since concentrations were not normally distributed based on the Kruskal-Wallis test (Kruskal and Walls, 1952). For each species, the Kruskal-Wallis one-way analysis of variance (ANOVA) on ranks among three periods was performed with pairwise comparison using Dunn's method (Dunn, 1964). (See track changes in Lines 201-207 in the revised manuscript).    Third, to test whether the use of data during 2000-2019 could bias annual trends of $PM_{2.5}$ and chemical components, we summarize measurement of $PM_{2.5}$ at a long-term monitoring site (in Quzhou County, North China Plain, operated by our group) during the period 2012-2020 from previous publications (Xu et al., 2016; Zhang et al., 2018, noted that data during 2017-2020 are unpublished before) (Figure S8). The results are consistent with trend in China from the meta-analysis (See track changes in lines 396-400 and lines 507-515 in the revised manuscript).

As suggested by the reviewer, we also improved trends analysis in $PM_{2.5}$ and its components to support the conclusion with same sites and long-term monitoring dataset. The three period are now compared using a non-parametric statistical method based on Kruskal-Wallis test. (See track changes in Lines 201-207 in the revised manuscript). Since the nationwide measurements of $PM_{2.5}$ and associated chemical components are lacking in 2010 in China, we newly add the information that we have undertaken a validation of CMAQ and its components (such as $SO_4^{2-}$, $NO_3^-$, and $NH_4^+$) using

available multi-observation datasets, including monitoring data at single site and satellite observations at regional scale (see track changes in Lines 539-590 in the revised manuscript and Figs. S3-S7 in the Supplementary Materials). Below we provide a point-by-point response to the reviewer's comments and how we have addressed them (including the line numbers for the track changes in the revised manuscript).

**Major comments:**

1.When analyzing the trends of $PM_{2.5}$ and SIA components using measurements collected from literature, the number of sites differ by a factor of four through the three periods. This makes me concern about the reliability of the trends reported in this study. I think the authors should use the same sites for trend analysis to keep consistency.

**Response:** Thank you for pointing this out. We now realize that the trend analysis in our study has some uncertainties. The historic trend analysis at the same sites were limited due to lack of long-term in situ measurements. In order to reduce the uncertainty of trend analysis, we have made some improvement in data analysis in the revised paper, as follows:

First, we re-filtered the data for meta-analysis and then made a three-period comparison using the measurements at sites that include both $PM_{2.5}$ and secondary inorganic ions ($SO_4^{2-}$, $NO_3^-$, and $NH_4^+$) (See tack changes in lines 298-304 in the revised manuscripts and updated Fig. 2). Second, our statistical analysis on the concentrations of $PM_{2.5}$ and secondary inorganic ions for three periods now uses a non-parametric statistical method since concentrations were not normally distributed based on the Kruskal-Wallis test

(Kruskal and Walls, 1952). For each species, the Kruskal-Wallis one-way analysis of variance (ANOVA) on ranks among three periods was performed with pairwise comparison using Dunn's method (Dunn, 1964). (See track changes in Lines 201-207 in the revised manuscript). Third, to test whether the use of data during 2000-2019 could bias annual trends of $PM_{2.5}$ and chemical components, we summarize measurement of $PM_{2.5}$ at a long-term monitoring site (in Quzhou County, North China Plain, operated by our group) during the period 2012-2020 from previous publications (Xu et al., 2016; Zhang et al., 2021, noted that data during 2017-2020 are unpublished before) (Fig. S8). The results are consistent with trend in China from the meta-analysis (See track changes in lines 396-400 and lines 507-515 in the revised manuscript).

2.Also, when discussing the trends based on meta-analysis, the authors left a lot of key questions unexplained. For example, why did the sulfate concentrations during hazy days increased from period I to period II while a series of $SO_2$ control policies has been implemented? Why did the nitrate concentrations not respond to the air pollution control policies from 2000 to 2019? In addition, an in-depth analysis about the drivers of the trends is lacking. The current manuscript just simply relates the trends with air pollution control policies and did not provide any quantitative analysis on the contributions from emission changes and meteorological impacts given that meteorological impacts can be much larger than the impacts from emission reductions (Sulaymon et al., 2021).

**Response:** The aim of our study is the analysis of trends in annual mean concentrations of $PM_{2.5}$, and chemical components, and SIA gaseous precursor, which help us to

identify responses for reduction of SIA and $PM_{2.5}$ pollution. We agree that the concentrations of $SO_4^{2-}$, $NO_3^-$ and $NH_4^+$ are influenced by meteorology as well as air quality policy but the impact of air quality policy manifests through longer-term trend whilst meteorology manifests as interannual variation. Our study focus is on investigating the temporal association between levels of SIA pollution and implementations of air quality policy. Before 2010, the Chinese government mainly focused on controlling $SO_2$ emission via improvement of energy efficiency. The 12th Five-Year Plan (2011-2016) added a reduction target for $NO_x$, but still with no attention paid to $NH_3$ abatement. The change of secondary inorganic aerosols (SIA, the sum of sulfate ($SO_4^{2-}$), nitrate ($NO_3^-$), and ammonium ($NH_4^+$)) were directly affected by these precursors ($SO_2$, $NO_x$, and $NH_3$). To confirm the contribution of precursor emission changes, not meteorological impacts, we undertook sensitivity analysis to analyze the SIA changes from 2010 to 2017 in four megacity cluster of eastern China under fixed meteorological condition (2010). We found that SIA show the downward trend (See Fig 6), which supports the SIA contribution to $PM_{2.5}$.

3. For all the simulations in this study, the authors did not provide any evaluation against measurements. Especially for the base simulations in sensitivity calculation, you need to first evaluate your simulated chemical regime in the SIA formation before you are conducting the $NH_3$/NOx/$SO_2$ emission reduction experiments and calculating the sensitivities of SIA ($PM_{2.5}$) formation to precursors emission changes. So you need to first evaluate your simulated sulfate, nitrate, ammonium, $SO_2$, $NO_2$, and $NH_3$ using measurements.

Response: We have now undertaken an extensive validation of CMAQ modelling

concentrations of $PM_{2.5}$ and its major components for January 2010 using surface measurements collected from publications and satellite observations. See the following new text (and associated new figures) in lines 543-590 in the revised manuscript for the presentation of this model validation.

"Since nationwide measurements of $PM_{2.5}$ and associated chemical components are lacking in 2010 in China, we undertook our own validation of $PM_{2.5}$ and its components (such as $SO_4^{2-}$, $NO_3^-$, and $NH_4^+$) using a multi-observation dataset that includes those monitoring data and satellite observations at a regional scale that were available.

First, the simulated monthly mean $PM_{2.5}$ concentration in January 2010 was compared with corresponding data obtained from the Tracking Air pollution in China (TAP, http://tapdata.org.cn/) database. The satellite historical $PM_{2.5}$ predictions are reliable (average $R^2 = 0.80$ and RMSE = 11.26 μg m$^{-3}$) in a validation against the in-situ surface observations on a monthly basis (Wei et al., 2020, 2021). The model well the captured spatial distributions of $PM_{2.5}$ concentrations in our studied regions of BTH, YRD, PRD, and SCB (Fig. S3a), with correlation coefficient ($R$) between simulated and satellite observed $PM_{2.5}$ concentrations of 0.96, 0.80, 0.60, and 0.85 for BTH, YRD, PRD, and SCB, respectively.

Second, we also collected ground-based observations from previous publications (Xiao et al., 2020, 2021; Geng et al., 2019; Xue et al., 2019) to validate the modeling concentrations of $SO_4^{2-}$, $NO_3^-$, and $NH_4^+$. Detailed information about the monitoring sites is presented in Table S5. The distributions of the simulated monthly mean concentrations of $SO_4^{2-}$, $NO_3^-$ and $NH_4^+$ in January 2010 over China is compared with collected surface measurements are shown in Fig. S4a, b, and c, respectively, with their

linear regression analysis presented in Fig. S4d. The model showed underestimation in simulating $SO_4^{2-}$ and $NO_3^-$ in the BTH region, which might be caused by the uncertainty in the emission inventory. The lack of heterogeneous pathways for $SO_4^{2-}$ formation in the CMAQ model might also be an important reason for the negative bias between simulations and measurements (Yu et al., 2005; Cheng et al., 2016). The model overestimated $NO_3^-$ concentration in the SCB region, but can capture the spatial distribution of $NO_3^-$ in other regions. The overestimation if $NO_3^-$ has been a common problem in regional chemical transport models such as CMAQ, GEOS-CHEM and CAMx (Yu et al., 2005; Fountoukis et al., 2011; Zhang et al., 2012; Wang et al., 2013c), due to the difficulties in correctly capturing the gas and aerosol-phase nitrate partitioning (Yu et al., 2005). The modeling of $NH_4^+$ concentrations show good agreement with the observed values. Generally, the evaluation results indicate that the model reasonably predicted concentrations of $SO_4^{2-}$, $NO_3^-$, and $NH_4^+$ in $PM_{2.5}$.

Third, we performed a comparison of the time-series of the observed and simulated hourly $PM_{2.5}$ and its precursors ($SO_2$ and $NO_2$) during January 2010. The model well captures the temporal variations of the $PM_{2.5}$ in Beijing, with an NMB value of 0.05 ug m$^{-3}$, NME of 28%, and $R$ of 0.92 (Fig. 5a). The predicted daily concentrations of $NO_2$ and $SO_2$ during January 2010 also show good agreement with the ground measurements in Beijing, with NMB and $R$ values of 0.12 ug m$^{-3}$ and 0.89 for $NO_2$, and -0.04, 0.95 for $SO_2$, respectively (Fig. 5b). The variations of daily $PM_{2.5}$ concentrations between simulation and observation at 4 monitoring sites (Shangdianzi, Chengdu, Institute of Atmospheric Physics, Chinese Academy of Sciences (IAP-CAS), and Tianjing) from 14 to 30 January 2010 also matched well, with NMB values ranging from -0.05 to 0.12 ug m$^{-3}$, and $R$ values exceeding 0.89 (Fig. S5c)."

4.The authors examined the trends of SIA and PM$_{2.5}$ based on observations collected from literature and explored the efficiency of NH$_3$ and acidic gases emission reduction using model simulations. However, they didn't build any connection between these two parts. They actually can use the observations to evaluate the simulated chemical regime before calculating the emission reduction efficiency. Or they can use model simulations to explore the drivers of the trends in the observed SIA and PM$_{2.5}$ concentrations through the three periods.

**Response:** The aim of this study is analysis of the trends of secondary inorganic aerosols and evidence for options to reduce SIA and PM$_{2.5}$ pollution. We believe the following methodology that we employed in our work should be clear in our manuscript. The contribution of SIA to PM$_{2.5}$ pollution was derived from assessment of observation data. We combined a meta-analysis and monitoring data to assess the difference in PM$_{2.5}$ and its chemical components between hazy and non-hazy days, which helps identify the major contributors to elevated PM$_{2.5}$. We also analysed the trend of PM$_{2.5}$ and its secondary inorganic aerosol precursors (SO$_2$, NO$_2$, and NH$_3$) during 2000-2019. This dataset derived from surface measurements and satellite observations. The potential of SIA and PM$_{2.5}$ concentration reduction from precursors emission reduction was simulated by the WRF-CMAQ model, which supports identification of options to reduce SIA and PM$_{2.5}$ pollution.

5.Also, while the meta-analysis shows that the nitrate concentrations do not significantly respond to air pollution control policies, the SIA sensitivity simulations show large decreases when reducing acidic gases emissions. Here I think the authors need to check whether the simulated nitrate concentrations decrease or not when

reducing NOx emissions and see if they are consistent with the observed nitrate concentration changes.

**Response:** We thanks the reviewer for their advice. In our sensitivity scenarios, the nitrate concentrations decreased with the reduction of $SO_2$ or $NO_x$ emissions. Previous studies also showed that $NO_x$ emissions control was important in mitigating nitrate pollution, and that SIA concentrations would decrease if $NO_x$ emission was reduced (Wang et al., 2013a, b). Li et al. (2021) showed that a 50% reduction in $NO_x$ emissions resulted in a 10.3% decrease in nitrate concentration in the BTH region in the winter of 2019. In addition, the validity of the chemical regime in the WRF-CMAQ model had been confirmed by our previous studies (Wang et al. 2020a, 2021b). The differing nitrate concentration between meta-analysis and model simulation may be explained that the model sensitivity scenarios of 2010, 2014, and 2017 are under fixed meteorological condition in order to identify the effectiveness of emissions reduction control and avoid the influence of meteorology.

6.Some of the references are not appropriate and do support their text.

**Response:** We have undertaken a full article check to ensure that we cite references that are relevant to our study. For instance, we corrected the references to Zhang et al. (2020) to support 50% $NH_3$ emission reduction in lines 264-266, and we have corrected the Liu et al 2019a and Liu et al., 2019b in lines 868-877 in the revised manuscript.

**Specific comments:**

7.Fig. 2: what does n represent? number of sites? The number of sites for the three periods differ by a factor of four (e.g. 93 vs 25 in Fig. 2 (a))? I think you need to use the same sites through the three periods to analyze the trends.

**Response:** Yes, *n* represents the number of sites. We now realize that the trend analysis in our study has some uncertainties. Please see our response above to comment #1, which deals with the same point, for full details of our revisions in respect of this comment.

8.Fig. 2: add "observed" or "measured" before "concentrations" in line 277 to make it clear that these data are measurements, not simulations. Same for Fig.3.

**Response:** We thank the reviewer for this suggestion. We have now added "observed" before "concentration" in Figs 2 and 3 to emphasize that these data are from measurements (See track changes in line 317 and lines 330-331 in the revised manuscript).

9.Also in Fig. 3, when you analyze the trends for each region, you need to use the same sites through the five years.

**Response**: Yes, our trend analysis uses the same sites for each region. The real-time monitoring data for $PM_{2.5}$, and $SO_2$ and $NO_2$ gaseous precursors to SIA, at 1498 monitoring stations in 367 cities during 2015-2019 were obtained from the China National Environmental Monitoring Center (CNEMC) (http://106.37.208.233:20035/). The $PM_{2.5}$, $SO_2$, and $NO_2$ trends for 2015-2019 in four mega-city clusters (BTH: Beijing-Tianjin-Hebei, YRD: Yangtze River Delta, SCB: Sichuan Basin, PRD: Pearl River Delta) used the same sets of sites.

10.Fig.3: why do you skip the years before 2015 given that you analyze trends from 2000 to 2019 in Fig. 2?

**Response:** The data shown in Fig. 3 were acquired from a large network operated by the China National Environmental Monitoring Center (CNEMC) (http://106.37.208.233:20035/). This network was initially built in 2013, in which the numbers of monitoring sites gradually increased and fully covered 367 cities in China since 2015 (1498 in situ sites). Therefore, to accurately access the annual trend, we selected the years before 2015. Fortunately, the period 2015-2019 covers the periods II and III that we define for air quality policy measures. Therefore, although this time periods of measurements are relatively short, it is still sufficient to investigate the trends in surface pollutant concentrations during period II and period III.

11. Line 264-266: How significant is the decreasing trend of 19.9%? Also, both the $PM_{2.5}$ concentrations during hazy and non-hazy days increased from period I to period II, which contradicts with line 270-271 and Fig. S2. What caused the increases in $PM_{2.5}$ concentrations between period I and period II?

**Response:** We thank the reviewer for pointing out this error. We have now corrected the sentence. The $PM_{2.5}$ concentrations from the literature review of hazy versus non-hazy days shows no significant change from period I and period II based on Kruskal-Wallis test. The observed $PM_{2.5}$ concentration has a decreasing trend from period I to period III, which is consistent with Fig 3d. This can be explained by $PM_{2.5}$ concentration responded positively to air policy implementations in China. (See track changes in Lines 298-304 in the revised manuscript).

13. Line 422: did you reduce NOx and $SO_2$ emissions by 50% simultaneously?

**Response:** Yes, we reduced the $NO_x$ and $SO_2$ emissions by 50% simultaneously. The sensitivity analysis aims to confirm the importance of acid gas emissions. So, we made

comparison between 50% reduction in NH$_3$ emissions and 50% reductions in acid gas (NO$_x$ and SO$_2$) emissions.

14.Line 305: what do you mean by 46 groups of data? do you mean data from 46 sites, including both measurement during hazy and non-hazy periods?

**Response:** The following text has been added to clarify what is meant. (See track changes in Line 350-354 in the revised manuscripts). "The 46 groups refer to independent analyses from the literature that compare concentrations of PM$_{2.5}$ and major components (SO$_4^{2-}$, NO$_3^-$, NH$_4^+$, OC, and EC) on hazy and non-hazy days measured across different sets of sites."

15.Fig.4 (A): what are the numbers on the right of the error bars? The number of sites?

**Response:** The numbers on the right of the error bars represents independent study sites. We added the information about the number on the right of error bars: "The *n* represents independent sites; more detail on this is presented in Section 2.2" (See track changes in Lines 380381 in the revised manuscript).

16.Line 306-313: what's the cause for the changes? meteorology (e.g. wind, precipitation), emissions or chemistry? I think here you need to consider the weather condition when you classify hazy or non-hazy days.

**Response:** In our meta-analysis study, the designation of a hazy or non-hazy day follows that used in the screened articles that are included. If the screened article did not use a designation of a hazy day, then days with PM$_{2.5}$ concentrations >75 μg m$^{-3}$

(the Chinese Ambient Air Quality Standard Grade II for $PM_{2.5}$ (CSC, 2012)) were treated as hazy days.

To avoid the influence of weather condition, we also used the WRF-CMAQ model to investigate the history of $PM_{2.5}$ and SIA concentration changes under fixed meteorological conditions (2010). This modelling approach supports the conclusion that secondary inorganic aerosols were the dominant contributor to ambient $PM_{2.5}$ concentrations.

17.Line 308-313 contradict with line 313-317: while your data shows no significant difference in the SIA portion (36-40%) between hazy and non-hazy days, you conclude SIA is the dominant role in haze pollution? In addition, in Fig 4. (B), 'other' plus OC is greater than 50%. What is 'other' in Fig. 4 (B)?

**Response:** Although the difference is not great (as the reviewer points out) it is nevertheless the case that the proportion of SIA components is higher on hazy days compared with non-hazy days. As we have noted in responses above, compared with other components the increase in concentrations was more significant (average increase of 98%) for the secondary inorganic ions $SO_4^{2-}$, $NO_3^-$, and $NH_4^+$ (see Figs 4A and 5). The "other" includes $Cl^-$, $F^-$, $Na^+$, $Ca^{2+}$, and $Mg^{2+}$. These other species are included in Fig 4A.

18.Fig. 5: again, if you want to compare the metrics from different periods, you need to use measurements from the same sites to keep consistency. Here, the number of sites for nitrate differ by a factor of 4. Also, the range of the x axis should be the same for these three plots for comparison.

**Response:** As already noted above, the historic trend analysis at the same sites was

limited due to lack of long-term in situ measurements. In order to reduce the uncertainty of trend analysis, we have made some improvement in the data analysis in three ways. For details, please see our response to the comment #4. In Fig. 5, we want to show the variations in PM$_{2.5}$ and its composition in different periods. We agree that comparisons across the different periods using the same sites would be better, but this work lacks data including PM$_{2.5}$ and its components at same sites. Therefore, we choose the "effect size" approach to assess the variation of PM$_{2.5}$ and its components between hazy days and non-hazy. The effect sizes were developed to normalize the combined studies outcomes to same scale. This was done through the use log response ratios. The variations in PM$_{2.5}$ and its components were evaluated in Meta-analysis of PM$_{2.5}$ and its chemical components in Section 2.

19.Line 332: what do you mean by "effect values"?

**Response:** The effect values were developed to normalize the combined studies' outcomes to the same scale. In our study this was done through the use of log response ratios (lnRR) (Nakagawa et al., 2012; Ying et al., 2019). The variations in aerosol species were evaluated as follows:

$$\ln RR = \ln \left(\frac{X_p}{X_n}\right)$$

(1)

where Xp and Xn represent the mean values of the studied variables of PM$_{2.5}$ components on hazy and non-hazy days, respectively. The mean response ratio was then estimated as:

$$RR = \exp \left[\sum \ln RR(i) \times W(i) / \sum W(i) \right]$$

(2)

where W(i) is the weight given to that observation as described below. Finally, variablerelated effects were expressed as percent changes, calculated as $(RR-1) \times 100\%$. A 95% confidence interval not overlapping with zero indicates that the difference is significant. A positive or negative percentage value indicates an increase or decrease in the response variables, respectively. We also used inverse sampling variances to weight the observed effect size (RR) in the meta-analysis to reduce the uncertainty from the number of studies. The effect values were evaluated in Meta-analysis of $PM_{2.5}$ and its chemical components in Section 2.

20. Line 335-338: 19.9% decrease (in average or in the median value?) from which period to which period? 49.6% decrease from which period to which period? Did you check the meteorology change (e.g. wind, precipitation, etc.) during the three periods? How can you make sure it's the $SO_2$ control policy not the meteorology change that caused the decrease in sulfate? Also, how do you explain the increase of sulfate during hazy days from period I to period II while you claim the $SO_2$ control policies were effective?

**Response:** We thank the reviewer for pointing out this mistake. We have corrected the sentence "Observed mean concentration of $SO_4^{2-}$ showed a downward trend from Period I to Period III on the non-hazy days and hazy days, decreasing by 38.6% and 48.3%, respectively" (See track changes in Line 388 in the revised manuscript). Both non-hazy days and hazy days show the downward trend. The difference of $SO_4^{2-}$ between hazy days and non-hazy days helps identify a reason for $PM_{2.5}$ formation. To confirm the decrease in sulfate was affected by $SO_2$ control policy we undertook the model sensitivity analysis of the trend of 2010, 2014, and 2017 under fixed meteorology. We found the sulfate showed downtrend trend (See Fig. 6).

21.Line 338-341 and line 350-351: So here do you mean that the NOx control policies since 2011 were not effective? If this is the case, how do you explain the difference between your conclusion and Fan et al. (2021), which reports decreasing trends in $NO_2$ observations in China from 2011 to 2019 owing to effective NOx control policies?

**Response:** We are sorry for confusing the reviewer. $NO_x$ control policies since 2011 were effective, which can be reflect by decreased $NO_x$ emissions and tropospheric $NO_2$ vertical column densities between 2011 and 2019 (Zheng et al., 2018; Fan et al., 2021). To avoid misunderstand, in the revised paper the mentioned sentences were revised as "In contrast, there were no significant downward trends in concentrations of $NO_3^-$ and $NH_4^+$ on either hazy or non-hazy days (Fig. 2c, d), but the mean $NO_3^-$ concentration in Period III decreased by 10.5% compared with that in Period II, especially on hazy days (-16.8%). These results could be partly supported by decreased $NO_x$ emissions and tropospheric $NO_2$ vertical column densities between 2011 and 2019 in China owing to effective $NO_x$ control policies (Zheng et al., 2018; Fan et al., 2021)."

22.Line 341-343: Kang et al. (2016) (Figure 1) shows a decreasing trend in Chinese total $NH_3$ emissions from 2000 to 2012 and doesn't show any further trends after 2012. How can this explain the 'the lack of downward trends in $NH_4^+$" in your Fig. 2d?

**Response**: Thank you for pointing this out. In the revised paper, we added a new reference (Liu et al., 2021) to support that the total $NH_3$ emission in China slightly increased between 2012 and 2018. Therefore, according to two references (Kang et al., 2016; Liu et al., 2021), the total $NH_3$ emission in China overall changed little and remained at high levels between 2000 and 2018, which could explain the lack of downward trends in particulate $NH_4^+$ found in our study.

In the revised paper, the mentioned sentences were revised as "The lack of significantly downward trends in $NH_4^+$ concentrations may be due to the fact that the total $NH_3$ emissions in China changed little and remained at high levels between 2000 and 2018, i.e., slightly decreased from 2000 (10.3 Tg) to 2012 (9.3 Tg) (Kang et al., 2016) and then slightly increased between 2013 and 2018 (Liu et al., 2021)."

23.Line 344-347: In Zhang et al. (2020) (the reference between line 818-821), I didn't find any data supporting your sentences here.

**Response:** We thank the reviewer for pointing this out. To make it clear, in the revised paper the mentioned sentences were revised as "The choice of 50% additional $NH_3$ emissions reduction is based on the feasibility and current upper bound of $NH_3$ emissions reduction expected to be realized in the near future (Liu et al., 2019a; Zhang et al., 2020; Table S4). For example, Zhang et al. (2020) found that the mitigation potential of $NH_3$ emissions from cropland production and livestock production in China can reach up to 52% and 58%, respectively."

24.Line 348-353: again, please make sure you are comparing the same sites for each region through these years.

**Response:** As noted above, our trend analysis uses the same sites for each region. The real-time monitoring data for $PM_{2.5}$, and $SO_2$ and $NO_2$ gaseous precursors to SIA, at 1498 monitoring stations in 367 cities during 2015-2019 were obtained from the China National Environmental Monitoring Center (CNEMC) (http://106.37.208.233:20035/). The $PM_{2.5}$, $SO_2$, and $NO_2$ trends for 2015-2019 in four mega-city clusters (BTH: Beijing-Tianjin-Hebei, YRD: Yangtze River Delta, SCB: Sichuan Basin, PRD: Pearl River Delta) used the same sets of sites.

25.Line 354-356: so what do you think is the reason that nitrate concentrations did not significantly respond to air pollution mitigation policies? Is it because NOx emissions did not really decrease? Or is it because the chemistry regime was actually $NH_3$ limited so that reducing NOx emission is not effective in reducing nitrate?

**Response:** The different trend between nitrate concentration and NOx emissions can be explained by the delayed response of emission reduction control. Before 2010, the Chinese government mainly focused on controlling $SO_2$ emission via improvement of energy efficiency, with less attention paid to NOx abatement. The 12[th] FYP (2011-2015) first added $NO_x$ regulation and required 10% reductions for $NO_x$. Some studies found that $SO_4^{2-}$ exhibited a much larger decline than $NO_3^-$ and $NH_4^+$, which led to a rapid transition from sulfate-driven to nitrate-driven aerosol pollution (Li et al.,2019). These transitions lead less change of $NO_3^-$ of SIA. The $NO_x$ and $NH_3$ emissions still have great potential for control in China.

26.Fig. 4 (A) and Fig. (5): how did you calculate the "variation"? Is it actually the ratio of the difference between concentrations during hazy and non-hazy days to the concentrations during non-hazy days?

**Response**: The variation was calculated through use of log response ratios (lnRR) which normalizes the combined studies outcomes to the same scale. We use this approach to calculate the difference of $PM_{2.5}$ and its component concentrations between hazy and non-hazy days. The calculation is described in Section 2.2 on Meta-analysis of $PM_{2.5}$ and its chemical components. Please also see our response to comment #19 for further details.

27.Line 358-363: Fig. 4 (B) (b) shows that ammonium and nitrate only account for 20-23% of total PM2.5 during both hazy and non-hazy days. And only 3% difference is found in their contribution (%) between hazy and non-hazy days. This seems to not support your sentences that nitrate and ammonium are currently a serious problem given that 'other' plus OC contribute more than 50% of total $PM_{2.5}$. Also, line 360, where is the sub figure (d) in Fig. 4 and Fig. 5?

**Response:** The SIA is identified as a major influencing factor on $PM_{2.5}$ for two reasons. First, the SIA components are the largest single component of $PM_{2.5}$, comprising 40%. All the other types of $PM_{2.5}$ component contribute considerably less than the SIA components. The "other" category incorporates all of $Cl^-$, $F^-$, $Na^+$, $Ca^{2+}$, and $Mg^{2+}$. Secondly, the SIA components are significantly higher on haze days compared with non-haze days than are the other components (See Figs 4A and 5). The sub figure (d) in Fig. 5 has been corrected by the sub figure (b) in Fig. 4 (See track changes in Lines 404 in the revised manuscript).

28.Figure S4: there seems to be a large bias in your simulated wind speed? Can you calculate the normalized mean bias for the comparisons?

**Response**: We thank the reviewer for pointing out a mistake in our data analysis of wind speed. After correcting the mistake, the $R$ and NMB values are 0.64 and 22.3% between the simulated and observed wind speed. We have added the MB, NMB and $R$ values inside the scatter plots of simulated versus observed T2, RH and wind speed (See track changes in lines 529-538 in the revised manuscript and Fig S7).

29.Fig. 6: Your simulated SIA concentrations over BTH are lower than those over YRD from 2010 to 2020. Have you evaluated your simulations (sulfate, nitrate, ammonium, total $PM_{2.5}$) using measurements?

**Response:** As the decreases of SIA concentration were obtained from model sensitivity experiments with the same meteorological conditions, we can't compare the simulated SIA using measurements. Our simulated SIA changes in the BTH region (2 ug m$^{-3}$ (equivalent to 6.8%) are consistent with other model simulations. For example, Ye et al. (2019) found that the annual average concentrations of $PM_{2.5}$, $SO_4^{2-}$, $NO_3^-$ and $NH_4^+$ in the BTH were reduced by 5.7%, 2.9-6.9%, 3.5-17.9%, and 4.2-23.3%, respectively, when agricultural $NH_3$ emissions were cut by 46.63%. Liu et al. (2021) also investigated that when $NH_3$ emissions in North China were reduced by 60%, the monthly mean population-weighted $PM_{2.5}$ concentrations in the BTH region decreased by 8.1 ug m$^{-3}$ (6.2%) in January 2015.

30.Line 376-378: I don't see any significant decreases in simulated SIA concentrations over BTH from 2010 to 2017 in your simulations without $NH_3$ emissions reductions (Fig. 6). Also, did you evaluate your simulated trends of SIA and $PM_{2.5}$ using measurements?

**Response:** The decreases of SIA and $PM_{2.5}$ were obtained from the sensitivity experiments with the same meteorological conditions (2010) so we can't compare the simulated trends of SIA and $PM_{2.5}$ using measurements. Our simulated SIA changes in the BTH (2 ug m$^{-3}$ (equivalent to 6.8%)) are consistent with other model simulations. For example, Ye et al. (2019) found that the annual average concentrations of $PM_{2.5}$, $SO_4^{2-}$, $NO_3^-$ and $NH_4^+$ in the BTH were reduced by 5.7%, 2.9-6.9%, 3.5-17.9%, and 4.2-

23.3%, respectively, when agricultural $NH_3$ emissions were cut by 46.63%. Liu et al. (2021) also investigated that when $NH_3$ emissions in North China were reduced by 60%, the monthly mean population-weighted $PM_{2.5}$ concentrations in the BTH region decreased by 8.1 ug m$^{-3}$ (6.2%) in January 2015. In addition, the $PM_{2.5}$ concentration from 2010 to 2017 were compared using one-way analysis of variance (ANOVA) test. There are significant decreases in simulated SIA concentrations over the BTH region from 2010 to 2017 in our simulations without $NH_3$ emissions reductions.

31.Line 379-380: why? Is it because that Sichuan has larger air pollutants emission reductions than PRD? Did you check the meteorology change? Most importantly, did you evaluate these using measurements?

**Response:** This comment refers to the following statement: "Across the four megacity clusters, the reduction in SIA and $PM_{2.5}$ is largest in the SCB region from 2010 to 2017 and smallest in the PRD region". The reductions in SIA and $PM_{2.5}$ referred to here are for the sensitivity simulations in 2014, and 2017 that used uniform pollutant ($NH_3$ or $NOx+SO_2$) emissions reductions and fixed 2010 meteorology. Therefore, meteorological impacts are not a factor in these data. According to the $PM_{2.5}$ observations obtained from the TAP database, the $PM_{2.5}$ concentration in the SCB region was much higher than that in the PRD region. Therefore, it is not surprising that the decreases of SIA and $PM_{2.5}$ concentrations in the SCB region were higher than that in the PRD region.

32.Line 384-385: I think the percentage reductions in simulated $PM_{2.5}$ is much smaller than those in SIA.

**Response:** Yes, the percentage reductions in simulated $PM_{2.5}$ is smaller than those SIA. We have now added the following sentence about reductions in $PM_{2.5}$ concentration: "The reductions of $PM_{2.5}$ in 2010, 2014 and 2017 were $9.7 \pm 0.1\%$, $9.0 \pm 0.1\%$, and $9.2 \pm 0.2\%$ in the megacity clusters, respectively." (See track changes in lines 443-445 in the revised manuscript).

33.Line 446-459: The $PM_{2.5}$ dataset from STET model are not real "observations". In addition, your $PM_{2.5}$ simulation show significant bias compared to the STET data. You need to evaluate your simulated SIA components and $SO_2/NO_2/NH_3$ using real observations and see if your simulated chemical regime is close to the true state or not. You already collected so many observations of SIA components, which can be used to evaluate your SIA simulations. Also, $SO_2/NO_2/NH_3$ observations are available from multiple satellite instruments.

**Response**: We agree with the reviewer. In the revised paper, we collected ground-based observations from the literature to verify the performance of the model of $PM_{2.5}$ and its chemical compositions with three approaches. These approaches are summarized again below, but for full details on the new model evaluations please see our response to comment #3.

First, the simulated monthly mean $PM_{2.5}$ concentration in January 2010 was compared with corresponding data from obtained from TAP database.

Second, the distribution of simulated monthly mean concentration of $SO_4^{2-}$, $NO_3^-$ and $NH_4^+$ in January 2010 over China compared with surface measurements are shown in Fig. S4a, b, and c, respectively, with their linear regression analysis showing in Fig. S4d.

Third, we performed a comparison of the time series of the observed and simulated hourly PM$_{2.5}$ and its precursors (SO$_2$ and NO$_2$) during January 2010.

34.Line 547-549: It seems that Fig. 2 (a) only show small decreases of PM$_{2.5}$ from 2000 to 2019 during non-hazy days, and no significant decreases were found during hazy days. Most importantly, the trends here are not reliable because the number of sites in your trend analysis differ by a factor of four.

**Response:** Thanks again for your suggestions. In order to reduce the uncertainty of trend analysis we have made some improvement in data analysis in the revised paper in the following three ways. First, we re-filtered the data for meta-analysis and then made a three-period comparison using the measurements at sites that include both PM$_{2.5}$ and secondary inorganic ions (SO$_4^{2-}$, NO$_3^-$, and NH$_4^+$) (See track changes in lines 298-304 in the revised manuscripts and updated Fig. 2). Second, the statistical analysis on the concentrations of PM$_{2.5}$ and secondary inorganic ions for three periods is replaced by using non-parametric statistical method since concentrations were not normally distributed based on Kruskal-Wallis test (Kruskal and Walls, 1952). (See track changes in Lines 201-209 in the revised manuscript). Third, we summarize measurement of PM$_{2.5}$ at long-term monitoring site (in Quzhou County, North China Plain, operated by our group) during the period 2012-2020 from previous publications (Xu et al., 2016; Zhang et al., 2021, noted that data during 2017-2020 are unpublished before). For full details on these improvements please see our response to comment #1.

35.Line 551-559: again, without any evaluation based on measurements of nitrate, sulfate, ammonium, NH$_3$, SO$_2$, and NO$_2$, your sensitivity calculations here are not reliable.

**Response**: We have provided detail response to the same point in our responses to comments # 3 and 33. In this revised manuscript we have added description of our new evaluation of the CMAQ output for $PM_{2.5}$ and its $SO_4^{2-}$, $NO_3^-$, and $NH_4^+$ components using a multi-observation dataset that includes monitoring data at single sites and satellite observations at regional scale that were available for the model simulated time period.

**References:**

Cheng, Y.F., Zheng, G.A., Wei, C., Mu, Q., Zheng, B., Wang, Z.B., Gao, M., Zhang, Q., He, K.B., Carmichael, G., Poschl, U., and Su, H.: Reactive nitrogen chemistry in aerosol water as a source of sulfate during haze events in China, Sci. Adv., 2(12), https://doi.org/10.1126/sciadv.1601530, 2016.

Dunn, O.J.: Multiple comparisons using rank sums. Technometrics., 6(3), 241-252, 1964.

Fan, C., Li, Z., Li, Y., Dong, J., van der A, R., and de Leeuw, G.: Variability of $NO_2$ concentrations over China and effect on air quality derived from satellite and ground-based observations, Atmos. Chem. Phys., 21, 7723 – 7748, https://doi.org/10.5194/acp-21- 7723-2021, 2021.

Fountoukis, C., Racherla, P. N., Denier van der Gon, H. A. C., Polymeneas, P., Charalampidis, P. E., Pilinis, C., Wiedensohler, A., Dall'Osto, M., O'Dowd, C., and Pandis, S. N.: Evaluation of a three-dimensional chemical transport model (PMCAMx) in the European domain during the EUCAARI May 2008 campaign,

Atmos. Chem. Phys., 11, 10331–10347, https://doi.org/10.5194/acp- 11-10331-
2011, 2011.

Geng, G.N., Xiao, Q.Y., Zheng, Y.X., Tong, D., Zhang, Y.X., Zhang, X.Y., Zhang, Q.,
He, K.B., and Liu, Y.: Impact of China's Air Pollution Prevention and Control
Action Plan on $PM_{2.5}$ chemical composition over eastern China, Sci. China. Earth.
Sci, 62, 1872-1884, https://doi.org/10.1007/s11430-018-9353-x, 2019.

Kruskal, W.H. and Wallis, W.A.: Use of ranks in one-criterion variance analysis. J. Am.
Stat. Assoc., 47(260), 1952.

Li, X., Bei, N., Hu, B., Wu, J., Pan, Y., Wen, T., Liu, Z., Liu, L., Wang, R. and Li, G.:
Mitigating $NO_X$ emissions does not help alleviate wintertime particulate pollution
in Beijing-Tianjin-Hebei, China, Environ. Pollut., 279(X), https://
10.1016/j.envpol.2021.11693, 2021.

Liu, X.J., Sha, Z.P., Song, Y., Dong, H.M., Pan, Y.P., Gao, Z.L., Li, Y.E., Ma, L., Dong,
W.X., Hu, C.S., Wang, W.L., Wang, Y., Geng, H., Zheng, Y.H. and Gu, M.N.:
China's atmospheric ammonia emission characteristics, mitigation options and
policy recommendations, Res. Environ. Sci., 34, 149-157,
https://doi.org/10.13198/j.issn.1001-6929.2020.11.12, 2021.

Nakagawa, S., and Santos, E. S. A.: Methodological issues and advances in biological
meta-analysis, Evol. Ecol., 26, 1253-1274, https://doi.org/10.1007/s10682-012-
9555-5, 2012.

Sulaymon, I. D., Zhang, Y., Hopke, P. K., Hu, J., Zhang, Y., Li, L., Mei, X.D., Gong,
K.J., Shi, Z.H., Zhao, B., and Zhao, F.X.: Persistent high $PM_{2.5}$ pollution driven by
unfavorable meteorological conditions during the COVID-19 lockdown period in

the Beijing-Tianjin-Hebei region, China. Environ. Res, 198, 111186. https://doi.org/10.1016/j.envres.2021.111186, 2021.

Sulaymon, I.D., Zhang, Y., Hopke, P.K., Zhang, Y., Hua, J. and Mei, X.: COVID-19 pandemic in Wuhan: Ambient air quality and the relationships between criteria air pollutants and meteorological variables before, during, and after lockdown, Atmos. Res, 250, https://doi.org/10.1016/j.atmosres.2020.105362, 2021.

Wang, L., Chen, X., Zhang, Y., Li, M., Li, P., Jiang, L., Xia, Y., Li, Z., Li, J., Wang, L., Hou, T., Liu, W., Rosenfeld, D., Zhu, T., Zhang, Y., Chen, J., Wang, S., Huang, Y., Seinfeld, J. H. and Yu, S.: Switching to electric vehicles can lead to significant reductions of $PM_{2.5}$ and $NO_2$ across China, One. Earth., 4, 1037–1048, https://doi.org/10.1016/j.oneear.2021.06.008, 2021b.

Wang, L., Yu, S., Li, P., Chen, X., Li, Z., Zhang, Y., Li, M., Mehmood, K., Liu, W., Chai, T., Zhu, Y., Rosenfeld, D. and Seinfeld, J. H. Significant wintertime $PM_{2.5}$ mitigation in the Yangtze River Delta, China, from 2016 to 2019: observational constraints on anthropogenic emission controls, Atmos. Chem. Phys., 2, 14787–14800, https://doi.org/10.5194/acp-20-14787-2020, 2020a.

Wang, Y., Yao, L., Wang, L., Liu, Z., Ji, D., Tang, G., Zhang, J., Sun, Y., Hu, B. and Xin, J.: Mechanism for the formation of the January 2013 heavy haze pollution episode over central and eastern China, Sci. China. Earth. Sci., 57(1), 14–25, https://doi.org/10.1007/s11430-013-4773-4, 2013a.

Wang, Y., Zhang, Q. Q., He, K., Zhang, Q. and Chai, L.: Sulfate-nitrate-ammonium aerosols over China: Response to 2000-2015 emission changes of sulfur dioxide,

nitrogen oxides, and ammonia, Atmos. Chem. Phys., 13(5), 2635–2652, https://doi.org/10.5194/acp-13-2635-2013, 2013b.

Wang, Y., Zhang, Q. Q., He, K.B., Zhang, Q., and Chai, L.: Sulfate-nitrate-ammonium aerosols over China: Response to 2000-2015 emission changes of sulfur dioxide, nitrogen oxides, and ammonia, Atmos. Chem. Phys., 13(5), 2635–2652, https://doi.org/10.5194/acp-13-2635-2013, 2013c.

Wei, J., Li, Z. Q., Cribb, M., Huang, W., Xue, W.H., Sun, L., Guo, J. P., Peng, Y. R., Li, J., Lyapustin, A.: Improved 1 km resolution $PM_{2.5}$ estimates across China using enhanced space–time extremely randomized trees, Atmos. Chem. Phys., 20, 3273-3289, https://doi.org/10.5194/acp-20-3273-2020, 2020.

Wei, J., Li, Z. Q., Lyapustin, A., Sun, L., Peng, Y. R., Xue, W. H., Su, T. N., Cribb, M.: Reconstructing 1-km-resolution high-quality $PM_{2.5}$ data records from 2000 to 2018 in China: spatiotemporal variations and policy implications, Remote. Sens. Environ., 252, 112136, https://doi.org/10.1016/j.rse.2020.112136, 2021.

Xiao, Q.Y, Geng, G.N., Liang, F.C., Wang, X., Lv, Z., Lei, Y., Huang, X,M., Zhang, Q., Liu, Y., and He, K.B.: Changes in spatial patterns of $PM_{2.5}$ pollution in China 2000–2018: Impact of clean air policies, Environ. Int., 141, 105776, https://doi.org/10.1016/j.envint.2020.105776, 2020.

Xiao, Q.Y., Zheng, Y.X., Geng, G.N., Chen, C.H., Huang, X.M., Che, H.Z., Zhang, X.Y., He, K.B., and Zhang, Q.: Separating emission and meteorological contribution to $PM_{2.5}$ trends over East China during 2000–2018, Atmos. Chem. Phys., 21, 9475-9496, https:// 10.5194/acp-21-9475-2021, 2021.

Xu, W., Wu, Q.H., Liu, X.J., Tang, A.H., Dore, A.J. and Heal, M.R.: Characteristics of ammonia, acid gases, and $PM_{2.5}$ for three typical land-use types in the North China Plain, Environ. Sci. Pollut. R., 23, 1158-1172, https:// doi.org/10.1007/s11356-015-5648-3, 2016.

Xue, T., Liu, J., Zhang, Q., Geng, G.N., Zheng, Y.X., Tong, D., Liu, Z., Guan, D.B., Bo, Y., Zhu, T., He, K.B., and Hao, J.M.: Rapid improvement of PM2.5 pollution and associated health benefits in China during 2013–2017, Sci. China. Earth. Sci., 62, 1847-1856, https:// 10.1007/s11430-018-9348-2, 2019.

Ye, Z.L., Guo, X.R., Cheng, L., Cheng, S.Y., Chen, D.S., Wang, W.L. and Liu, B.: Reducing $PM_{2.5}$ and secondary inorganic aerosols by agricultural ammonia emission mitigation within the Beijing-Tianjin-Hebei region, China, Atmospheric. Environ., 219. https:// 10.1016/j.atmosenv.2019.116989, 2019.

Ying, H., Yin, Y. L., Zheng, H. F., Wang, Y. C., Zhang, Q. S., Xue, Y. F., Stefanovski, D., Cui, Z. L., and Dou, Z. X.: Newer and select maize, wheat, and rice varieties can help mitigate N footprint while producing more grain, Glob. Change. Biol., 12, 4273-4281, https://doi.org/10.1111/gcb.14798, 2019.

Yu, S.C., Dennis, R., Roselle, S., Nenes, A., Walker, J., Eder, B.,Schere, K., Swall, J., and Robarge, W.: An assessment of the ability of three-dimensional air quality models with current thermodynamic equilibrium models to predict aerosol $NO_3^-$, J. Geophys. Res-Atmos, 110(D7), https://doi.org/10.1029/2004JD004718, 2005.

Zhang, L., Jacob, D. J., Knipping, E. M., Kumar, N., Munger, J. W., Carouge, C. C., van Donkelaar, A., Wang, Y. X., and Chen, D.: Nitrogen deposition to the United

States: distribution, sources, and processes, Atmos. Chem. Phys., 12, 4539–4554, https://doi.org/10.5194/acp-12-4539-2012, 2012.

Zhang, X. M., Gu, B. J., van Grinsven, H., Lam, S.K., Liang, X., Bai, M., and Chen, D.L.: Societal benefits of halving agricultural ammonia emissions in China far exceed the abatement costs, Nat. Commun., 11, 4357. https://doi.org/10.1038/s41467-020-18196-z, 2020.

Zhang, Y.Y., Liu, X.J., Zhang, L., Tang, A.H., Goulding, K. and Collett Jr, J.L.: Evolution of secondary inorganic aerosols amidst improving $PM_{2.5}$ air quality in the North China plain, Environ. Pollut., 281, 117027, https://doi.org/10.1016/j.envpol.2021.117027, 2021.

---

## Referee Report (RR1)

The author analyzed the long-term trends of PM$_{2.5}$ chemical components and their drivers using numerical models. Some issues still need to be addressed although the author have made great efforts according to previous comments. I suggest major revision for the manuscript prior to be finally published in ACP.

1. In the introduction, the author spent lots of length to describe the air pollutants control measures for SO$_2$, NO$_x$ and NH$_3$ during different stages in the past decade. Since the aims of this manuscript is to evaluate the changes of SIA as responses to stringent measures, more results of SIA variations should be cited and summarized. Indeed, extensive researches have reported on this topic. Also, the author should compare the trends in PM$_{2.5}$ and components in this paper with previous studies.

2. Considering that the author divided 2000-2019 into three periods, e.g., period I (2000-2012), period II (2013-2016) and period III (2017-2019), annual trend used in the analysis might not be appropriate. Annual trend often refers to year-to-year variations. Measurements during three periods covered different seasons (winter, summer etc.) and sites (urban, suburban or rural), these actually influenced the conclusions because it's well known that more polluted air quality frequently occurred during wintertime in urban site. The author used PM$_{2.5}$ at a long-term monitoring site during 2012-2020 to verify the decreasing trend summarized from meta-analysis. I supposed that this evidence could only support that the decreasing trend was reliable. The quantitative results, e.g., decreased by 8.2% from period I to period III, was still to be evaluated. It is a bit confused that the author collected publications covering four-season measurements to summary the trends from period I to period II, however, only January was chosen to do simulations. The author explained that severe haze pollution often occurred in January. The effectiveness of precursors controlling measures could be season-dependent. That's another uncertainty for this study.

3. The author attributed all the variations of PM$_{2.5}$ and chemical components to changes of gaseous precursors resulting from control measures. The reasons were somewhat pale and inadequate. In fact, the responses of SIA to precursors were complex and sometimes non-linear. Previous studies have concluded that enhanced atmospheric oxidation capacity, faster deposition of total inorganic nitrate and the changes of atmospheric circulation could be possible drivers. That's likely why PM$_{2.5}$ showed no significant trend from period I to period II despite control measures were implemented since 2013. Please cite more relevant publications, and then rephrase and expand the related explanations throughout the study. In current revised manuscript, the results and discussion were flat.

4. For model simulations, the author fixed meteorology in 2020 to exclude the impacts of meteorology. Thus, the CMAQ simulation before and during the COVID-lockdown didn't represent the actual results during these periods. In Figure S6, the author compared the CMAQ results with ground observations for PM$_{2.5}$, SO$_2$ and NO$_2$. This is not reasonable. The author should firstly do simulations using real

meteorology to evaluate the performance of CMAQ model, and then do controlled experiments using fix meteorology. Also for Figure S3-S5, the author only assessed the simulation results in January 2010 with observations. Indeed, they should do these year by year using real modelling results.

5.  The author compared the model results between 50% reductions in $NH_3$ emissions and 50% reductions in acid gases, concluding that reducing acid gases is more effective. Did the author do sensitivity cases with reductions of 50% $NH_3$ and 50% acid gases, which might be more close to the facts. Another issue is quantifying how much the precursors should be decreased to fulfill air quality targets.

6.  In Figure 4, all measurements were averaged to derive the two pie charts. The author only filtered the data for meta-analysis using the measurements at sites that include both $PM_{2.5}$ and SIA, we noticed that many of these re-filtered measurements didn't include mental species ($Na^+$, $Mg^{2+}$, $Ca^{2+}$, $F^-$), which called "Other", accounting for 36.8-37.4% during non-haze and haze days. The inconsistency among measurements used for averaging species caused large uncertainties to the conclusions. Studies simultaneously measured all species could be more reliable and scientific, at less for the pie charts.

7.  The author concluded that increased SIA formation is the major driving factor for haze pollution, which was obviously true consistent with previous studies. Due to the limitations of collecting datasets from publications instead of long-term filed measurements, the contribution of SIA slightly increased from 36% during non-haze days to 40% during haze days. The concentrations of SIA and other $PM_{2.5}$ components synchronously increased from non-haze to haze days. Thus, it is not appropriate and convincing to draw this conclusion solely based on this study.

8.  The first reviewer mentioned that the results in Figure 2a and b,c,d crossed several pages, and the interruption makes it hard to read. In the response, the author only added more detail figure caption to Figure 2. Indeed, the reviewer suggested to recombine the figures, rephrase the sentences or rearrange the paragraphs, making them more coherent in the context.

9.  In Figure 7, S3, S4 and S7, the south China Sea were missed in maps. This is really less rigorous.

10. The author added more citations in the revised manuscript, which were not shown in the Reference.

---

## Author Response (AR2)

**Response letter** to reviewer comments on the manuscript "Trends in secondary inorganic aerosol pollution in China and its responses to emission controls of precursors in wintertime" by Fanlei Meng, Yibo Zhang, Jiahui Kang, Mathew R. Heal, Stefan Reis, Mengru Wang, Lei Liu, Kai Wang, Shaocai Yu, Pengfei Li, Jing Wei, Yong Hou, Ying Zhang, Xuejun Liu, Zhenling Cui, Wen Xu, Fusuo Zhang.

Response: We thank the reviewers for their comments, which have helped us substantially to improve our manuscript. Below, we explain how we incorporated the comments into the revised version. Our responses are given in blue below, and revisions to the manuscript are shown in track changes (with line number references).

**Reviewer#1**

The author analyzed the long-term trends of $PM_{2.5}$ chemical components and their drivers using numerical models. Some issues still need to be addressed although the author have made great efforts according to previous comments. I suggest major revision for the manuscript prior to be finally published in ACP.

1.In the introduction, the author spent lots of length to describe the air pollutants control measures for $SO_2$, $NO_x$ and $NH_3$ during different stages in the past decade. Since the aims of this manuscript is to evaluate the changes of SIA as responses to stringent measures, more results of SIA variations should be cited and summarized. Indeed, extensive researches have reported on this topic. Also, the author should compare the trends in $PM_{2.5}$ and components in this paper with previous studies.

Response: Thank you for your suggestions. In the introduction, we have added information about change of SIA to the introduction as follows: "Following the successful controls on $NO_x$ and $SO_2$ emission since 2013 in China, some studies found $SO_4^{2-}$ exhibited a much larger decline than $NO_3^-$ and $NH_4^+$, which led to a rapid transition from sulfate-driven to nitrate-driven aerosol pollution (Li et al., 2019, 2021; Zhang et al., 2019)." In the results we have added the following: "Li et al.(2021) also found that $SO_4^{2-}$ exhibited a significant decline, However, $NO_3^-$ did not evidently exhibit a decreasing trend in the BTH region". See track change in Lines 105-108 and Lines 406-408 in the revised manuscript.

2.Considering that the author divided 2000-2019 into three periods, e.g., period I (2000-2012), period II (2013-2016) and period III (2017-2019), annual trend used in the analysis might not be appropriate. Annual trend often refers to year-to-year variations. Measurements during three periods covered different seasons (winter, summer etc.) and sites (urban, suburban or rural), these actually influenced the conclusions because it's well known that more polluted air quality frequently occurred during wintertime in urban site. The author used $PM_{2.5}$ at a long-term monitoring site during 2012-2020 to verify the decreasing trend summarized from meta-analysis. I supposed that this evidence could only support that the decreasing trend was reliable. The quantitative results, e.g., decreased by 8.2% from period I to period III, was still to be evaluated. It is a bit confused that the author collected publications covering four-season measurements to summary the trends from period I to period II, however, only January was chosen to do simulations. The author explained that severe haze pollution often

occurred in January. The effectiveness of precursors controlling measures could be season-dependent. That's another uncertainty for this study.

**Response:** Thank you for these points. The period 2000-2019 was divided into three periods on the basis of China's emission control policies: period I (2000-2012), in which $PM_{2.5}$ was not the targeted pollutant; period II (2013-2016), the early stage of targeted $PM_{2.5}$ control policy implementation; and period III (2017-2019), the latter stage with more stringent policies. We agree that there can be variation in $PM_{2.5}$ between different seasons (winter, summer, etc) and site type (urban, suburban or rural). In the Uncertainty analysis and Limitations, we have added the following: "Considering the uncertainty of $PM_{2.5}$ and its major components between different seasons (winter, summer, etc) and site type (urban, suburban or rural). We have analyzed historic trend in the different season and sites (Figs. S13-S20). We found that concentrations of $PM_{2.5}$ and its major chemical components ($SO_4^{2-}$, $NO_3^-$, and $NH_4^+$) were significantly higher in Fall and Winter than in Spring and Summer (Fig. S13). Only the Winter season showed significant change trend in the three periods (Figs. S14-S17). The analyses also confirmed that pollution days predominated in Winter. We also found that concentrations of $PM_{2.5}$ and its major chemical components were higher at urban than rural sites (Fig.S18). Spatially, the trends of $PM_{2.5}$ and its major components are similar across the whole of China (both of urban and rural) (Fig.S19). Rural areas show the same change trend in hazy days compared with whole of China (Fig. S20)." See track change in Lines 513-524 in the revised manuscript and newly Figs. S13-S20 in the Supplementary Materials.

January was selected as the typical simulation month because wintertime haze pollution frequently occurs in this month (Wang et al., 2011; Liu et al., 2019b). January of 2010 was also found to have PM$_{2.5}$ pollution more serious than other months (Geng et al., 2017, 2021). Whilst we agree the effectiveness of precursors controlling measures could be season dependent, we chose winter for our case study for identifying the effective options to reduce PM$_{2.5}$ and SIA pollution because winter is always the most polluted time. We will explore the effectiveness of precursor emissions reductions in different seasons in future work. See track change in Lines 247-248 in the revised manuscript.

[Figure]

[revised manuscript text omitted]
 author attributed all the variations of $PM_{2.5}$ and chemical components to changes of gaseous precursors resulting from control measures. The reasons were somewhat pale and inadequate. In fact, the responses of SIA to precursors were complex and sometimes non-linear. Previous studies have concluded that enhanced atmospheric oxidation capacity, faster deposition of total inorganic nitrate and the changes of atmospheric circulation could be possible drivers. That's likely why $PM_{2.5}$ showed no significant trend from period I to period II despite control measures were implemented since 2013. Please cite more relevant publications, and then rephrase and expand the related explanations throughout the study. In current revised manuscript, the results and discussion were flat.

**Response:** Thank you for your suggestions. In the revised paper, we have summarized more references to explain the trends of $PM_{2.5}$ and SIA during the three periods: The $PM_{2.5}$ showed no significant trend from period I to period II despite control measures implemented since 2013. This can be explained by the enhanced atmospheric oxidation capacity (Huang et al., 2021), faster deposition of total inorganic nitrate (Zhai et al., 2021) and the changes of atmospheric circulation (Zheng et al., 2015; Li et al.,2020). See track changes in Lines 302-305 in the revised manuscript.

4.For model simulations, the author fixed meteorology in 2020 to exclude the impacts of meteorology. Thus, the CMAQ simulation before and during the COVID-lockdown didn't represent the actual results during these periods. In Figure S6, the author compared the CMAQ results with ground observations for $PM_{2.5}$, $SO_2$ and $NO_2$. This is not reasonable. The author should firstly do simulations using real meteorology to

evaluate the performance of CMAQ model, and then do controlled experiments using fix meteorology. Also for Figure S3-S5, the author only assessed the simulation results in January 2010 with observations. Indeed, they should do these year by year using real modelling results.

**Response:** We are sorry for confusing the reviewer. The simulation results in Figure S6 did use real, not fixed, meteorological conditions. The year-by-year evaluations using real modelling results is helpful to validate the reliability of the CMAQ model. Thank you for your suggestions. We also newly evaluated the model performance in actual meteorological conditions for $PM_{2.5}$ concentrations in January 2014 and 2017, respectively. As shown in the Figure S21, the model well captured the spatial distribution of $PM_{2.5}$ concentration in China with MB (NMB) values of 23.2 ug m$^{-3}$ (15.4%) and 26.8 ug m$^{-3}$ (-26.7%) for 2014 and 2017, respectively. The simulated $PM_{2.5}$ concentrations compared well against the observations, with $R$ values of 0.82 and 0.65, respectively. See track changes in lines 587 and lines 595-601 in the revised manuscript.

[Figure]

**Figure S21.** Overlay of observed (colored circles) and simulated (color map) monthly concentrations of $PM_{2.5}$ in January 2014 and 2017.

5. The author compared the model results between 50% reductions in $NH_3$ emissions and 50% reductions in acid gases, concluding that reducing acid gases is more effective. Did the author do sensitivity cases with reductions of 50% NH3 and 50%acid gases, which might be more close to the facts. Another issue is quantifying how much the precursors should be decreased to fulfill air quality targets.

**Response:** Yes, we did the sensitivity analysis with reduction of 50% $NH_3$ and 50% acid gases. The result was already shown in Fig 7 in the main manuscript (also reproduced again below). We found the reductions in SIA concentration are 13.4±0.5% greater for the 50% reductions in $SO_2$ and $NO_x$ emissions than for the 50% reductions in $NH_3$ emissions. We thank for reviewer's suggestions to quantify how much the precursors should be decreased to meet the air quality targets. The aim of our study is to analysis the trends of secondary inorganic aerosol and strategic options to reduce SIA and $PM_{2.5}$ pollution in China. This study focused on finding the effective options in terms of precursor gas emissions reductions. In a future study we will explore the suggestions to identify how much the precursors should be reduced to meet the air quality targets.

[Figure]

**Fig. 7.** Left: the spatial distributions of simulated $PM_{2.5}$ concentrations (in μg m-3) in January 2017 with (a) 50% reductions in ammonia ($NH_3$) emissions and (b) 50% reductions in acid gas ($NO_x$ and $SO_2$) emissions. Right: the % decreases in $PM_{2.5}$ (c) and SIA (d) concentrations for the simulations with compared to without the $NH_3$ and acid gas emissions reductions in four megacity clusters (BTH: Beijing-Tianjin-Hebei, YRD: Yangtze River Delta, SCB: Sichuan Basin, PRD: Pearl River Delta). ** denotes significant differences without and with 50% ammonia emission reductions (P <0.05). n is the number of calculated samples by grid extraction. Error bars are standard errors of means.

6.In Figure 4, all measurements were averaged to derive the two pie charts. The author only filtered the data for meta-analysis using the measurements at sites that include both $PM_{2.5}$ and SIA, we noticed that many of these re-filtered measurements didn't include

mental species (Na+, $Mg^{2+}$, $Ca^{2+}$, $F^-$), which called "Other", accounting for 36.8-37.4% during non-haze and haze days. The inconsistency among measurements used for averaging species caused large uncertainties to the conclusions. Studies simultaneously measured all species could be more reliable and scientific, at less for the pie charts.

**Response:** We thank the reviewer for pointing this out. We agree that studies that simultaneously measure all species would be better, but data that includes $PM_{2.5}$ and all its components at the same sites is incomplete. In our study, we filtered the data for meta-analysis using the measurements at sites that include both $PM_{2.5}$, OC, EC, and secondary inorganic ions ($SO_4^{2-}$, $NO_3^-$, and $NH_4^+$). The "Other" species was calculated by difference between $PM_{2.5}$ and sum of OC, EC, and secondary inorganic ions ($SO_4^{2-}$, $NO_3^-$ and $NH_4^+$). This approach can reduce the uncertainty in the difference of $PM_{2.5}$ and its chemical components on both hazy and non-hazy days. To make this clear, in the revised paper we added newly state that "The "Other" species was calculated by difference between $PM_{2.5}$ and sum of OC, EC, and secondary inorganic ions ($SO_4^{2-}$, $NO_3^-$ and $NH_4^+$)." See track changes in Lines 355-357 in the revised manuscript.

7.The author concluded that increased SIA formation is the major driving factor for haze pollution, which was obviously true consistent with previous studies. Due to the limitations of collecting datasets from publications instead of long-term filed measurements, the contribution of SIA slightly increased from 36% during non-haze days to 40% during haze days. The concentrations of SIA and other $PM_{2.5}$ components synchronously increased from non-haze to haze days. Thus, it is not appropriate and convincing to draw this conclusion solely based on this study.

**Response:** In response to this comment from the reviewer we have now removed from our manuscript the conclusion that increased SIA formation is the major driving factor for haze pollution. (See track changes in lines 41-42 in the revised manuscript).

8.The first reviewer mentioned that the results in Figure 2a and b,c,d crossed several pages, and the interruption makes it hard to read. In the response, the author only added more detail figure caption to Figure 2. Indeed, the reviewer suggested to recombine the figures, rephrase the sentences or rearrange the paragraphs, making them more coherent in the context.

**Response:** Thank you for your suggestions. In the revised paper, we have revised Fig 2. The aim of this figure is to show the trends in observed concentration of $PM_{2.5}$, $SO_4^{2-}$, $NO_3^-$, and $NH_4^+$ between non-hazy and hazy days in Period (2000-2012), Period II (2013-2016), and Period III (2017-2019). (See track changes in Line 317 in the revised manuscript).

[Figure]

**Fig. 2.** Comparisons of observed concentrations of (a) PM2.5, (b) SO42-, (c) NO3-, and (d) NH4+ between non-hazy and hazy days in Period I (2000–2012), Period II (2013–2016), and Period III (2017–2019). Bars with different letters denote significant differences among the three periods ($P<0.05$) (upper and lowercase letters for non-hazy and hazy days, respectively). The upper and lower boundaries of the boxes represent the 75th and 25th percentiles; the line within the box represents the median value; the whiskers above and below the boxes represent the 90th and 10th percentiles; the point within the box represents the mean value. Comparison of the pollutants among the three-periods using Kruskal-Wallis and Dunn's test. The n represents independent sites; more detail on this is presented in Section 2.2.

9. In Figure 7, S3, S4 and S7, the south China Sea were missed in maps. This is really

less rigorous.

**Response:** Thanks for reviewer's point this. We have corrected the maps of Figure 7, S3, S4, S7, S11, and S12.

[Figure]

**Fig. 7.** Left: the spatial distributions of simulated $PM_{2.5}$ concentrations (in μg m$^{-3}$) in January 2017 with (a) 50% reductions in ammonia ($NH_3$) emissions and (b) 50% reductions in acid gas ($NO_x$ and $SO_2$) emissions. Right: the % decreases in $PM_{2.5}$ (c) and SIA (d) concentrations for the simulations with compared to without the NH3 and acid gas emissions reductions in four megacity clusters (BTH: Beijing-Tianjin-Hebei, YRD: Yangtze River Delta, SCB: Sichuan Basin, PRD: Pearl River Delta). ** denotes significant differences without and with 50% ammonia emission reductions ($P<0.05$). n is the number of calculated samples by grid extraction. Error bars are standard errors of means.

[Figure]

**Figure S3.** (a) Simulated and observed monthly mean PM$_{2.5}$ concentrations (μg m$^{-3}$) for January 2010. The observations are from the China High Air Pollutants (CHAP, https://weijing-rs.github.io/product.html) database. (b) Scatter plots of simulated versus observed monthly means PM$_{2.5}$ concentration in the BTH, YRD, PRD, and SCB regions.

[Figure]

**Figure S4.** Overlay of observed (colored circles) and simulated (color map) monthly mean concentrations of (a) $SO_4^{2-}$, (b) $NO_3^-$ and (c) $NH_4^+$ in January 2010. (d) scatter plot of simulated and observed concentrations of $SO_4^{2-}$, $NO_3^-$ and $NH_4^+$. The dotted lines correspond to the 1:2 and 2:1 lines. The observations are collected from the literature (See Table S5).

[Figure]

**Figure S11**. The spatial distributions of simulated SIA concentrations (in μg m$^{-3}$) without (a) and with (b) 50% ammonia emissions reduction for the years 2010, 2014, 2017 and 2020. The % decreases in SIA concentrations in each year for the simulations with the emissions reductions are shown in row (c). (Period I (2000–2012), Period II (2013–2016), and Period III (2017–2019); Special control is the restrictions in economic activities and associated emissions during the COVID-19 lockdown period in 2020.)

[Figure]

**Figure S12.** The spatial distributions of simulated PM$_{2.5}$ concentrations (in μg m$^{-3}$) without (a) and with (b) 50% ammonia emissions reduction for the years 2010, 2014, 2017 and 2020. The % decreases in PM$_{2.5}$ concentrations in each year for the simulations with the emissions reductions are shown in row (c). (Period I (2000–2012), Period II (2013–2016), and Period III (2017–2019); Special control is the restrictions in economic activities and associated emissions during the COVID-19 lockdown period in 2020.)

10.The author added more citations in the revised manuscript, which were not shown in the Reference.

**Response**: Thank you for pointing this out. We have undertaken a full article check to ensure that we cite references that are relevant to our study. For instance, we corrected the references to Zhang et al. (2020b) to in lines 403. We have deleted Röllin et al., 2004 that was previously in lines 926-929 and Sulaymon et al., 2021 that was in lines 940-943.

**Reviewer# 2**

After reading the authors' response letter and the revised manuscript, it appears to me that the revision has adequately addressed the previous comments. In particular, the revised manuscript has reasonably evaluated the model simulations of air pollution with available measurements and as well included the continuous measurements of aerosol components at a surface site over 2012-2020 to support their results, addressing the major concerns in its previous version. The manuscript now presents sufficiently new information on how aerosol levels may respond to acid gas and ammonia emission reductions in China, and I suggest publish on ACP

One more comment is that most of the numbers presented in the manuscript are percentage values, while we may be also interested in the absolute concentration changes. I suggest the authors add one Table (e.g., in the Supplement) summarizing the values shown in Figure 6, so that the aerosol concentration changes at different emission scenarios are clear.

**Response:** We thanks the reviewer for their supportive comments on the substantial amendments we made to our manuscript at the previous revision and for their recommendation for publication in ACP. In response to their one additional comment, we have now added a new Table S6 in the Supplementary Materials to show the values corresponding to the values shown in Fig 6.

**Table S6** Simulated SIA concentrations ( in μg m$^{-3}$) with (basic) and 50% ammonia (NH$_3$) emissions reductions in January for years 2010, 2014, 2017, and 2020 in four megacity clusters.

| | 2010 (Period I) | | 2014 (Period II) | | 2017 (Period III) | | 2020 (Special control) | |
|---|---|---|---|---|---|---|---|---|
| | Base | 50%NH$_3$ | Base | 50%NH$_3$ | Base | 50%NH$_3$ | Base | 50%NH$_3$ |
| BTH | 29.9±1.2 | 24.0±1.1 | 29.9±1.2 | 24.4±1.1 | 27.8±1.1 | 23.1±1.0 | 21.6±0.8 | 19.6±0.8 |
| YRD | 42.7±0.9 | 31.6±0.8 | 41.5±0.9 | 31.1±0.8 | 37.8±0.9 | 28.8±0.8 | 26.9±0.5 | 22.6±0.5 |
| SCB | 57.8±1.2 | 43.5±1.1 | 52.9±1.0 | 41.4±1.0 | 44.5±0.8 | 35.9±0.8 | 28.8±0.5 | 25.2±0.5 |
| PRD | 13.9±0.5 | 10.0±0.3 | 11.9±0.4 | 8.7±0.3 | 10.3±0.4 | 7.5±0.3 | 7.2±0.2 | 5.9±0.2 |

Note: The value is mean ± standard errors of means. (Period I (2000–2012), Period II (2013–2016), and Period III (2017–2019); Special control is the restrictions in economic activities and associated emissions during the COVID-19 lockdown period in 2020. BTH: Beijing-Tianjin-Hebei, YRD: Yangtze River Delta, SCB: Sichuan Basin, PRD: Pearl River Delta).